# An intriguing new diapsid reptile with evidence of mandibulo-dental pathology from the early Permian of Oklahoma revealed by neutron tomography

Ethan D. Mooney[1,2], Tea Maho[1,2], Joseph J. Bevitt[3], Robert R. Reisz[1,2]*

1 International Center of Future Science, Dinosaur Evolution Research Center, Jilin University, Changchun, Jilin, Peoples Republic of China, 2 Department of Biology, University of Toronto at Mississauga, Mississauga, Ontario, Canada, 3 Australian Nuclear Science & Technology Organization, Australian Center Neutron Scattering, Lucas Heights, Sydney, New South Wales, Australia

* robert.reisz@mail.utoronto.ca

## Abstract

The initial stages of diapsid evolution, the clade that includes extant reptiles and the majority of extinct reptilian taxa, is surprisingly poorly known. Notwithstanding the hypothesis that varanopids are diapsids rather than synapsids, there are only four araeoscelidians and one neodiapsid present in the late Carboniferous and early Permian. Here we describe the fragmentary remains of a very unusual new amniote from the famous cave deposits near Richards Spur, Oklahoma, that we recognize as a diapsid reptile, readily distinguishable from all other early amniotes by the unique dentition and lower jaw anatomy. The teeth have an unusual reeding pattern on the crown (long parallel ridges with rounded surfaces), with some teeth posteriorly tilted and strongly recurved, while a ventral protuberance forms the anterior terminus of the dentary. Overall, the lower jaw is unusually slender with a flattened ventral surface formed by the dentary and splenial anteriorly and the angular in the mid-region. The presence of a very slender triradiate jugal revealed through computed tomography confirms the existence of a large lower temporal fenestra, while the medial edge of the maxilla and the anatomy of the palatine confirm the presence of a large suborbital fenestra. Computed tomography of this new taxon reveals maxillary innervation that is characteristically reptile, not synapsid. Although no other definitively identifiable skull roof elements exist, the suborbital fenestra borders preserved on the palatine and maxilla supports the hypothesis that this is a diapsid reptile. Interestingly, the right dentary shows evidence of pathology, a rarely reported occurrence in Paleozoic amniotes, with several empty tooth sockets filled by bone. This small predator with delicate subthecodont implanted dentition provides strong evidence that diapsid reptiles were already diversifying rapidly in the early Permian, but likely were relatively rare members of terrestrial vertebrate assemblages.

**Data Availability Statement:** Specimens are housed at the Royal Ontario Museum. ROMVP 87366, holotype, anterior region of skull and mandibles. CT segmented. ROMVP 87367, right

dentary with resorption pits visible on lingual side. CT segmented. ROMVP 87368, right dentary with resorption pits visible for two teeth in positions d08 and d09. ROMVP 87369, right dentary with two replacement teeth visible in positions d14 and d16. ROMVP 87370, left dentary, anterior region for tooth positions d01 to d03. Specimen was histologically examined. CT data can be provided upon request from the Department of Natural History, Royal Ontario Museum. CT data were also uploaded to Morphobank. It is now active. https://morphobank.org/index.php/Projects/ProjectOverview/project_id/4395.

**Funding:** Supported by a Natural Sciences and Engineering Research Council (NSERC) scholarship to T.M., and NSERC grant to R.R.R., and the Jilin University, China. The funders had no role in study design, data collection and analysis, decision to publish, or preparation of the manuscript.

**Competing interests:** The authors have declared that no competing interests exist.

# Introduction

## Taxonomic background

The first appearance of amniotes in the fossil record are represented by fragmentary remains of at least two taxa in the late Carboniferous *Sigillaria* stumps of Joggins, Nova Scotia [1, 2], now dated at 318 Ma [3]. These include the reptile *Hylonomus* and the synapsid *Protoclepsydrops*. Although other amniotes represented by various basal eureptiles, parareptiles, and numerous synapsids have been discovered in the Carboniferous and early Permian, diapsids remained very rare in the Paleozoic fossil record [4–6]. Diapsids represent an extremely diverse clade of amniotes that includes all extant reptiles as well as birds and crocodiles, but also a great diversity of extinct taxa. Diapsid reptiles dominated the Mesozoic Era, and even recovered after the end Cretaceous extinction event to include the very successful and diverse clades of birds, turtles and squamates. Diapsids first show up in the fossil record only near the end of the Carboniferous (304 Ma) [7, 8] and remain poorly represented in the fossil record of the early Permian.

The earliest known diapsid reptile, *Petrolacosaurus* from the Upper Pennsylvanian Rock Lake Shale of Kansas [7], already was characterized by the presence of a suborbital fenestra, bound by the palatine, and maxilla [8], and two temporal fenestrae, consisting of a lower or inferior temporal fenestra bound by the jugal, quadratojugal, postorbital, and squamosal, and an upper or superior temporal fenestra bound by the postorbital, parietal, and squamosal. Younger diapsids include other basal araeoscelids [9], the neodiapsid *Orovenator* [4] from the Permo-Carboniferous of North America, and a handful of late Permian taxa from western Europe, Russia, Madagascar, and South Africa [10–16]. It is in this context that we describe here a new diapsid reptile from the early Permian of Oklahoma.

## Locality background

The Dolese Brothers Limestone Quarry near Richards Spur, Oklahoma, has yielded an unprecedented number of fossils and the most diverse terrestrial Paleozoic assemblage known to date, consisting of hundreds of thousands of specimens and more than 40 tetrapod taxa [17]. This locality dates to the early Permian between 289–286 Ma and is an exceptional representation of the rarely preserved upland environment, in contrast to the far more common floodplain, lowland preservational settings [17–19]. Only two other early Permian localities in the world preserve terrestrial upland Paleozoic faunal assemblages, the Bromacker Quarry in Germany [20] and the Bally Mountain locality in Oklahoma [21, 22], neither of which match the richness of the Richards Spur locality. The faunal assemblage at Richards Spur represents various amniotes and anamniotes, many being endemic to this location [19]. The rarest amniotes at this locality are the diapsids *Orovenator mayorum* and an unnamed diapsid (based on a single parietal), both of which appear to be endemic to this locality [1, 4]. *Orovenator mayorum* is the oldest known neodiapsid and is the only other well-articulated diapsid described from this locality [4]. Two specimens of *Orovenator mayorum* have been recovered, both partial skulls [4]. It has been suggested by Reisz et al. [4] that the lack of diapsids can be attributed to ecological partitioning involving the separation of diapsids into araeoscelidians and neodiapsids to lowland and upland habitats respectively.

The fossiliferous deposits composed of early Permian infills are limited to an interconnected system of karst fissures within the surrounding Ordovician limestone [17]. These fissures contain vast numbers of disarticulated tetrapod remains, and much rarer articulated and semi-articulated skeletons, indicating that the infilling of these caves occurred frequently with

surface sediment containing skeletal remains during wash events like severe rainstorms [17, 18]. During the early Permian, this area would have been subject to monsoonal conditions and fluctuations in seasonality brought on by the climatic conditions associated with being in equatorial Southwestern Laurasia [23, 24].

## Significance

In this study we describe a new diapsid from the Richards Spur locality in Oklahoma through the utilization of high-resolution neutron micro-computed tomography (μCT) data to create a comprehensive 3D rendering, exposing the skull elements in remarkable detail. This new small mysterious amniote is a new taxon, distinct from all other known members of this assemblage and unlike any other amniote from the early Permian. This discovery provides valuable new information about diapsid evolution in the early Permian and enriches our understanding of early diapsids in the Paleozoic Era.

## Materials and methods

### Neutron tomography

Tomographic data for specimen ROMVP 87366 was acquired using the thermal-neutron imaging instrument DINGO situated at and tangentially facing the 20 MW Open-Pool Australian Lightwater (OPAL) reactor housed at the Australian Nuclear Science and Technology Organisation (ANSTO), Lucas Heights, New South Wales, Australia. Neutron tomography is a non-destructive process that achieves exceptional contrast for specimens from Richards Spur, owing to the transparency of the largely limestone matrix to neutrons, and the relatively high attenuation of neutrons by the naturally occurring hydrocarbons present within the embedded bones.

For this study, DINGO was configured in high-flux mode, with a collimation ratio ($L/D$) of 500, where $L$ is the neutron aperture-to-sample length and $D$ is the neutron aperture diameter, to yield a flux at sample of $4.75 \times 10^7$ n·cm$^{-2}$s$^{-1}$ [25]. A Teledyne Photometrics Iris 15™ large field of view scientific cMOS camera (16-bit, $5056 \times 2960$ pixels) was used, coupled with a Makro Planar 100 mm Carl Zeiss lens and a 20 μm thick terbium-doped Gadox scintillator screen (Gd$_2$O$_2$S:Tb, RC Tritec AG) to yield a pixel size of $13.7 \times 13.7$ μm and field of view was of 69 x 40 mm$^2$.

The tomographic scan consisted of a total of 1440 equally-spaced angle shadow-radiographs obtained every 0.125˚ as the sample was rotated 180˚ about its vertical axis, with the specimen positioned 17 mm from the detector face. Both dark (closed shutter) and beam profile (open shutter) images were obtained for calibration before initiating shadow-radiograph acquisition. To reduce anomalous noise, a total of three individual radiographs with an exposure length of 14s were acquired at each angle [26]. Total scan time was 19 hours.

The individual radiographs were summed in post-acquisition processing using the Grouped ZProjector function, and anomalous white-spots removed using a threshold filter in ImageJ v.1.51h. Normalisation and tomographic reconstruction of the 16-bit raw data were performed using Octopus Reconstruction v.8.8 (Inside Matters NV), yielding virtual slices perpendicular to the rotation axis.

Neutron radioactivation of the specimen was recorded 30 min post-scan using a Canberra Radiagem Survey Meter Dosimeter dose, yielding a 30s average dose rate on contact of 63 mSv/h. After a period of 14 days, no residual radioactivity was recorded, and the specimen was cleared for return.

## Histology

Prior to histological analysis, all specimens were photographed using Leica DVM6 digital microscope and LAS X software, registered to R. R. Reisz at the University of Toronto Mississauga.

The fragmented left dentary (ROMVP 87370) was individually embedded in Castolite AC polyester resin, placed in the vacuum and then left to cure for approximately 24-hours. Later the specimen was cut, either longitudinally or cross-sectionally, using the Metcut-5 low-speed saw (MetLab) with a diamond wafer blade at 225 rpm. The specimens were mounted on frosted plexiglass slides and recut using the Metcut-5 saw in order to later grind the specimens using the Metcut-10 Geo (MetLab) machine with a grinding cup. Then the slides were manually ground with a progressively finer grit, between 1000- to 2000-grit, silicon carbide paper. Lastly, the thin-sectioned specimens were imaged using a Nikon DS-Fi1 camera mounted onto a Nikon AZ-100 microscope using NIS Elements-Basic Research software registered to R. R. Reisz of the University of Toronto Mississauga.

The replacement period was examined for one functional tooth of ROMVP 87370 containing a resorption pit on the lingual side of the jawbone where the replacement tooth would have been developing [27, 28]. In order to determine the age of the functional tooth, the incremental lines of von Ebner within the dentine were counted, starting from the pulp cavity and continuing perpendicularly towards the exterior edge of the tooth. The age of the missing replacement tooth was done by calculating an estimate for the height of the replacement tooth, which was determined to be two-thirds of the height of the resorption pit of the functional tooth. This was then overlayed on the exterior edge of the functional tooth apex and the incremental lines of von Ebner within the area were counted. The replacement period was determined by taking the difference between the age of the functional tooth age and the estimated age of the replacement tooth [27, 29].

## Nomenclatural acts

"The electronic edition of this article conforms to the requirements of the amended International Code of Zoological Nomenclature, and hence the new names contained herein are available under that Code from the electronic edition of this article. This published work and the nomenclatural acts it contains have been registered in ZooBank, the online registration system for the ICZN. The ZooBank LSIDs (Life Science Identifiers) can be resolved and the associated information viewed through any standard web browser by appending the LSID to the prefix "https://can01.safelinks.protection.outlook.com/?url=http%3A%2F%2Fzoobank.org%2F&data=05%7C01%7Crobert.reisz%40utoronto.ca%7Cd97a3dc3a47f44f90c9f08dabbf96cda%7C78aac2262f034b4d9037b46d56c55210%7C0%7C0%7C638028977364866541%7CUnknown%7CTWFpbGZsb3d8eyJWIjoiMC4wLjAwMDAiLCJQIjoiV2luMzIiLCJBTiI6Ik1haWwiLCJXVCI6Mn0%3D%7C3000%7C%7C%7C&sdata=rsST4QLtDFacXwh0TWmaI8c92lduy%2BRXe8D8oVtSIr4%3D&reserved=0". The LSID for this publication is: urn: lsid: zoobank.org:pub: 42833F34-1F4E-4938-858C-BF9E7B1A964C. The electronic edition of this work was published in a journal with an ISSN, and has been archived and is available from the following digital repositories: PubMed Central, LOCKSS.

## Systematic paleontology

Reptilia Laurenti, 1768

Eureptilia Olson 1947

Diapsida Osborn, 1902

Neodiapsida Benton 1985

*Maiothisavros* gen. nov.
 urn: lsid: zoobank.org:act: 644A2B53-7AA7-4DED-8F5A-OBE0729AA1CC

*Maiothisavros dianeae* sp. nov
 urn: lsid: zoobank.org:act: CB710B2D-03B6-46CD-89C1-78535516F697

*Maiothisavros = Μάιος (Máios)* + thisavrós, Greek, "treasure") *dianeae*

The generic name from Classical Greek *maio* represents 'May', in honour of Bill May, arguably the greatest contributor to the collection of fossils from Dolese Brothers Limestone Quarry near Richards Spur, and the Greek *thisavros* for 'treasure'.

 The species name *dianeae*, honours Diane Scott for her dedication to the field of paleontology over her more than 4-decade long career.

## Material

ROMVP 87366, holotype, anterior region of skull and mandible. CT segmented.

ROMVP 87367, right dentary with resorption pits visible on lingual side. CT segmented.

ROMVP 87368, right dentary with resorption pits visible for two teeth in positions 8 and 9.

ROMVP 87369, right dentary with two replacement teeth visible in positions 14 and 16.

ROMVP 87370, left dentary, anterior region for tooth positions 1 to 3. Specimen was histologically examined.

## Locality

Fossiliferous fissure fill deposits within limestone karsts at Richards Spur, OK, USA. These deposits represent a continental upland environment dating to the early Permian (Cisuralian), indicating an early Artinskian age of the fauna [17].

## Diagnosis

Small diapsid eureptile characterized by the presence of strongly curved marginal dentition displaying a reeded pattern of the enamel surface of the crown (long parallel ridges with rounded surfaces), narrow mandible anteriorly, and ventrally expanded anterior dentary terminus to accommodate well developed symphysis. The flattened, rather than rounded, ventral surface of the mandible is formed by the dentary, splenial, and angular bones. The slender triradiate jugal has narrow suborbital and subtemporal rami that differs from mycterosaurine and varanodontine varanopids in lacking any lateral suborbital ornamentation. The teeth on the transverse flange of the pterygoid are nearly as large as the dentition on the dentary. The presence of an anterior coronoid as well as what may be a fragmentary posterior coronoid are also rare among diapsids.

## Description

The holotype specimen ROMVP 87366 is fragmentary, with much of the skull and the posterior half of the mandible missing (Fig 1). The relative position of the various elements suggests

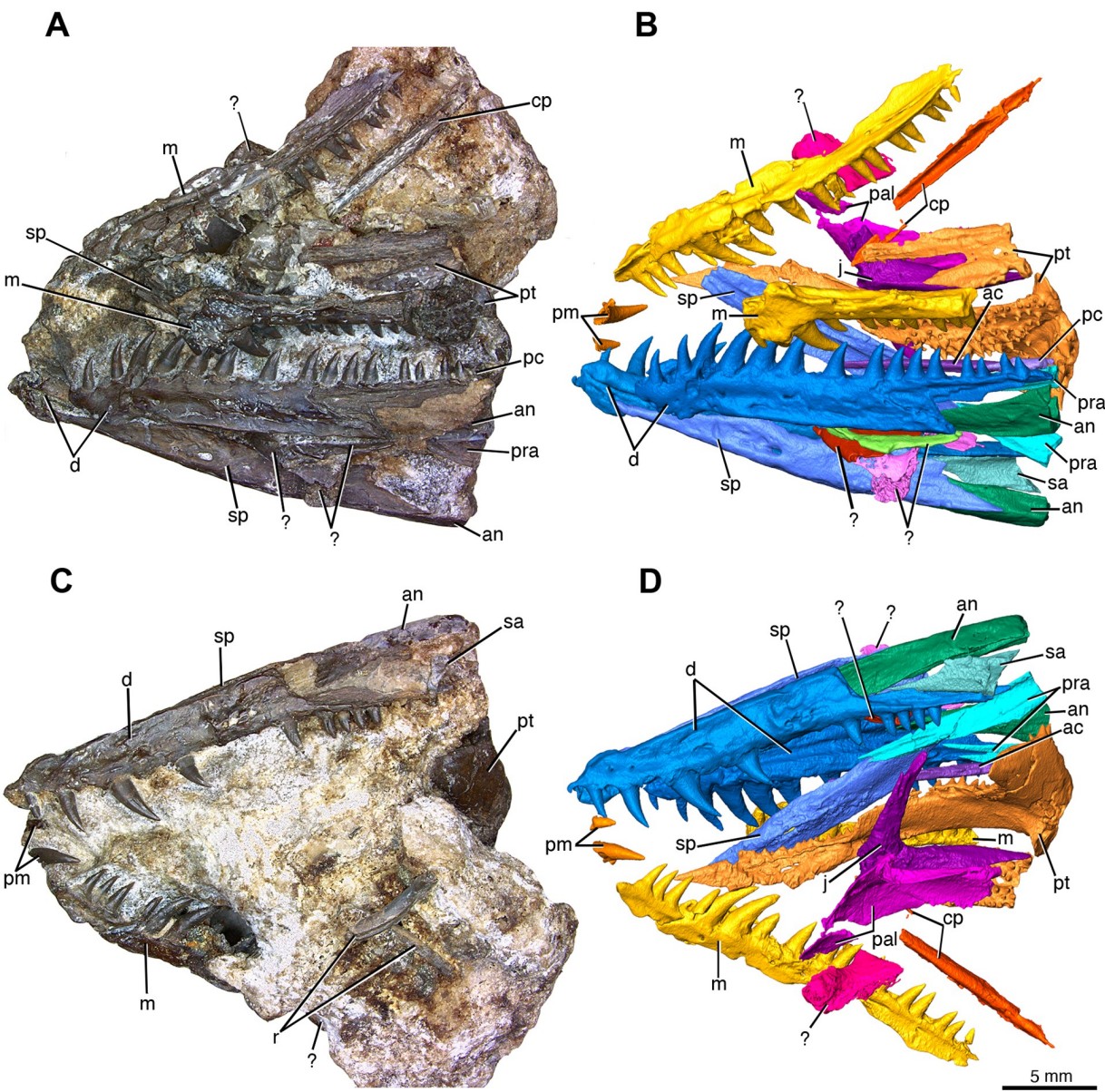

**Fig 1. Partial skull of *Maiothisavros dianeae* (ROMVP 87366) and corresponding renderings based on CT imaging sequences.** (A) left lateral view and (C) right lateral view of *Maiothisavros dianeae* ROMVP87366 specimen photo. (B) left lateral and (D) right lateral three-dimensional model of *Maiothisavros dianeae* from CT image sequences. Abbreviation: an, angular; ac, anterior coronoid; cp, cultriform process; d, dentary; j, jugal; m, maxilla; pal, palatine; pc, posterior coronoid; pm, premaxilla; pra, prearticular; pt, pterygoid; r, rib; sa, surangular; sp, splenial;?, unknown element.

that this was once a complete or near-complete skull that partially disarticulated, and only a portion of the original fossil was recovered. Most of the elements belong to the mandible and the palate, as well as a partial premaxillary dentition, the maxillae, and the right jugal. The lower jaw elements are preserved in a partially articulated position and appear to have a slight concave curvature of the lateral surface most noticeable in both dentaries. Two disarticulated fragmentary ribs and a few amorphous bone fragments were also present in ROMVP 87366, but do not provide any diagnostic information and are consequently omitted from this description. Most bones from ROMVP 87366 are well preserved. The dentition is

comparatively large relative to the size of the skull, which was likely less than 6 cm in length. The overall shape of the skull is also very gracile, and the shape of the mandible suggests that the skull was unusually slender, at least anteriorly. The general labiolingual thickness of the existing dentition and posteriorly angled tooth crowns suggests a capture-and-kill predatory behavior [30].

## Skull roof

The premaxilla is only represented by 2 seemingly in-situ teeth spatially associated with the right maxilla (Fig 1). The single space between the two teeth is mesiodistally equivalent to the other premaxilla teeth, suggesting at least three teeth would have been present on the premaxilla. These teeth are sharply pointed, conical, recurved, and are labiolingually wide at the base with a narrowing towards the crown apex. The premaxillary teeth appear to lack serrations on the medially trending carinae of the tooth crown. The first tooth appears to be much smaller, compared to the posterior-most premaxillary tooth, which is approximately twice as large in both tooth height and labiolingual basal diameter. Although this conforms with the small size of the first dentary tooth, the first premaxillary teeth may be lacking most of the crown base. The posterior premaxillary tooth also appears to have a greater degree of posterior recurvature than the directly adjacent anterior maxillary teeth and its apex displays a greater degree of medial curvature. The premaxillary dentition appears to display a surficial texture consistent with that found in the anterior dentary and maxillary dentition, they are conical with a labiolingually wide base, slightly recurved, display mesial and distal carinae, and possess plicidentine surrounding the dentine base.

The maxilla is a mediolaterally narrow, elongated element displaying at least 24 tooth positions (Fig 1) but may have had more since the posterior portion of each maxilla is missing. Nearly the entire dorsal portion of both maxillae are visibly missing, as evidenced by the fractured dorsal surface. The right maxilla contains a tooth count of 24, whereas the less complete left maxilla contains 13 teeth. The maxillae also display characteristic subthecodont teeth that are conical, recurved, labiolingually wide at the base, and have carinae on the mesial and distal cutting edges with plicidentine surrounding the dentine base (coronated vertical striations are elaborated on in the tooth attachment and implantation section). Exceptionally prominent carinae are present on the large caniniform teeth 6 and 7, curving laterally from the apex down the length of the tooth and are followed posteriorly by a groove (Fig 2). The lateral surface of the maxilla is smooth with few scattered foramina throughout and a prominent lateral protuberance directly above the caniniform teeth, suggesting that there was a well-developed dorsal process. The medial surface displays a few large foramina and a supramaxillary foramen that opens into channelized grooves merging into the smooth dorsomedial ridge extending the length of the maxilla. The medial surface of the alveolar shelf is smooth and rounded anteriorly for the lateral border of the internal naris, and posteriorly for the suborbital fenestra, typically found in diapsid reptiles.

Computed tomography (CT) in conjunction with visual observations reveal that the main trunk of the maxillary canal is tubular, much thicker, and longer than the other projections of the maxillary canal (Fig 3). The main trunk begins at the anterior-most portion of the maxilla and runs parallel to the long axis of the maxilla in the anterior-posterior direction just above the maxillary dentition. On the lateral surface of the right dentary small external foramina projections branch from the main canal at tooth positions 6, 9, 11, 14, and an incomplete projection at position 16. The maxillary canal also becomes exposed posteriorly after tooth 15, where it is then directed by a prominent lingual groove in the anterior-posterior direction. Five external foramina projections are also visible on the lateral surface of the left maxilla at tooth

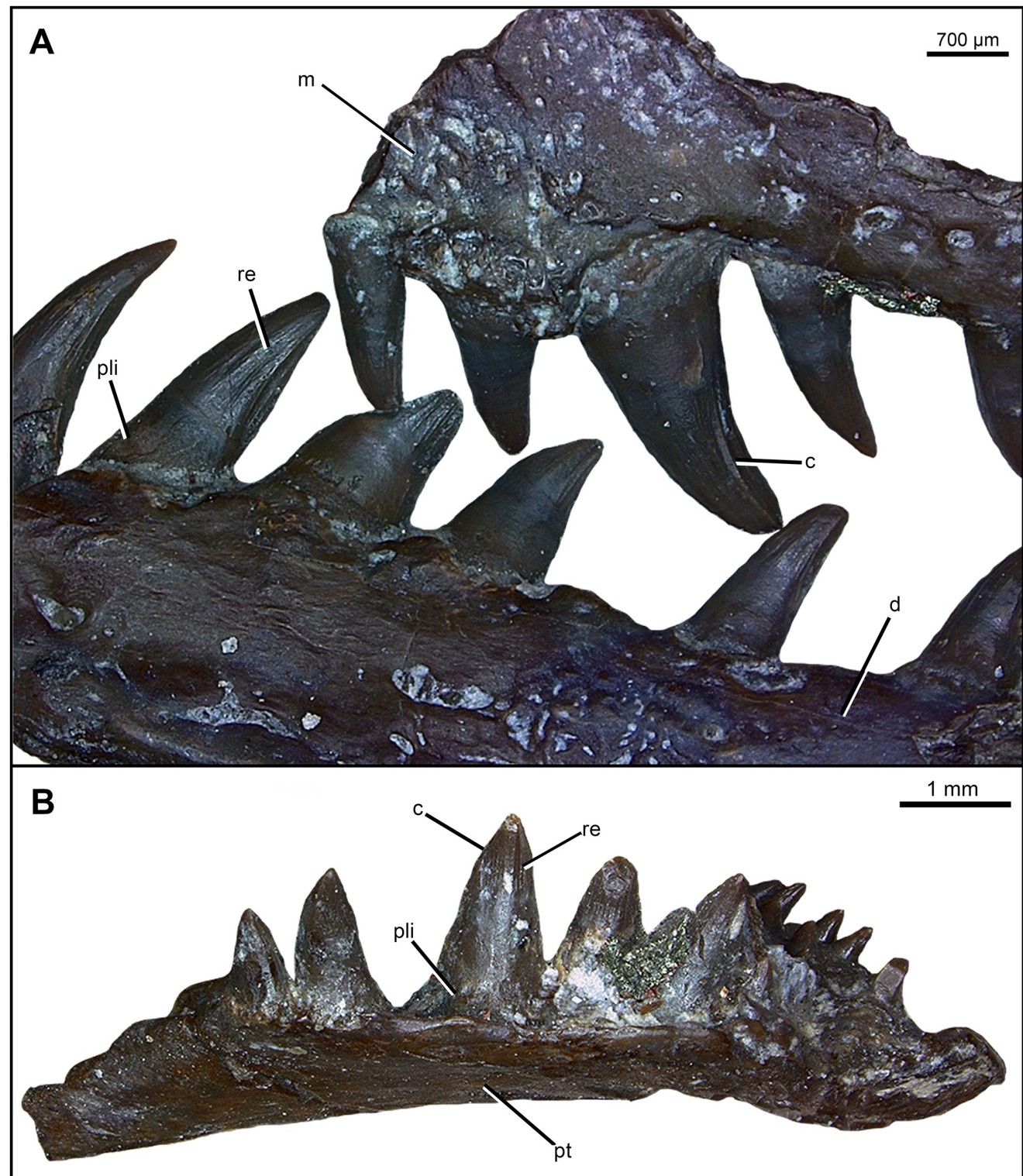

**Fig 2. Specimen photograph displaying vertical reeding on the tooth crown of dentary, maxilla, and transverse flange of the pterygoid in *Maiothisavros dianeae* (ROMVP 87366).** (A) left dentary and maxilla in lateral view, (B) transverse flange of the pterygoid in posterior view. Abbreviation: c, carina; d, dentary; re, reeding; m, maxilla; pli, plicidentine; pt, pterygoid.

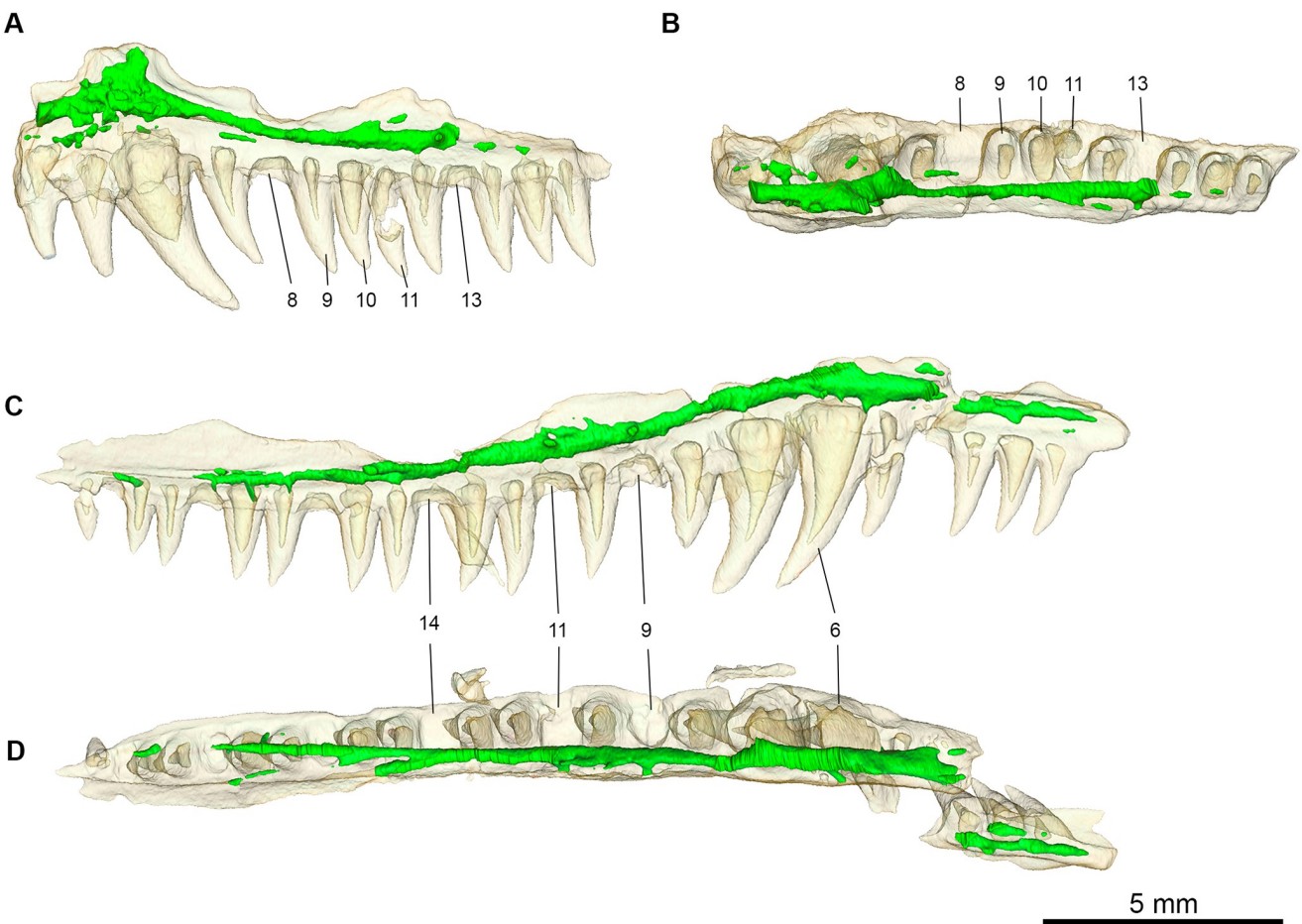

**Fig 3. Rendering of maxillary innervation from CT imaging sequences of *Maiothisavros dianeae* (ROMVP 87366).** Innervation shown in green for (A) left maxilla in lateral view and (B) and dorsal view, (C) right maxilla in lateral view and (D) dorsal view.

positions 8, 9, 10, 11, and 13. The maxillary canal is most easily viewed in lateral view as the tubular shape appears to be mediolaterally compressed. The dorsal portion of this projection is not preserved; however, the ventral portion is displayed in the left maxilla. The dorsal projection of the maxillary canal is rather wide at the ventral most portion and indicates an extension in the dorsal direction. There are also notably few smaller lateral foraminal projections radiating from the main maxillary canal. Larger lateral foraminal projections occur in the more posterior portion of the maxilla, whereas relatively more frequent smaller lateral foraminal projections are concentrated in the anterior portion of the maxilla.

Although incomplete, the right jugal (Fig 4) exhibits the triradiate shape typical of early diapsids. The dorsal process is complete and displays a characteristic anteriorly concave spoonlike shape up to the tip where the anterior surface would suture to the postorbital bone. This dorsal process is not vertical but rather tilted posteriorly to form an acute angle with the posterior subtemporal process. The dorsal process has a well-developed medial ridge, making it much wider mediolaterally than anteroposteriorly, and appears to have a significant contribution to the orbital margin along its slightly concave anterior surface and to the temporal fenestra along its slightly convex posterior surface. The anterior suborbital process is robust but is broken anteriorly. The posterior subtemporal process is bifurcate posteriorly and elongated with a free ventral surface for much of its length where the jugal forms the ventral border of

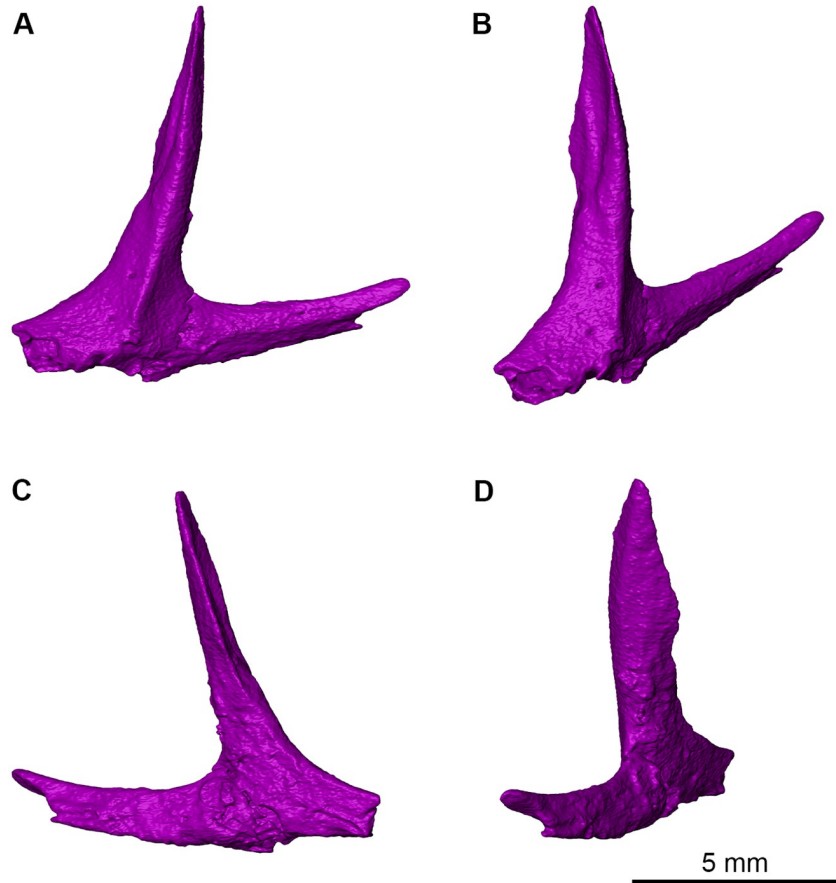

**Fig 4. Rendering of *Maiothisavros dianeae* (ROMVP 87366) right jugal from CT image sequences.** (A) and (B) are medial views, whereas (C) and (D) are lateral views of jugal from ROMVP 87366.

the skull, as well as a short posterior extension that would have overlain the quadratojugal. It also appears that the maxilla reaches the level of the orbital bar or midpoint of the jugal and would not directly contact the quadratojugal. The subtemporal process of the jugal is relatively slender and rod-like, unlike the broad sheet-like condition seen in araeosceloid diapsids, or the more robust process in derived varanopids, and is more like those seen in neodiapsids. The external surface of the bone is well-preserved and lacks the kind of rugosity seen in some varanopids, like *Mesenosaurus*, a common synapsid at the locality [31]. The medial surface is smooth and displays two small foramina on the anterior side of the ventrodorsally extending ridge. Due to poor preservation, the lateral surface is rough and clearly displays a large fracture. Overall, the jugal of *Maiothisavros* appears to be much more robust than that of the neodiapsid *Orovenator*, and in many respects similar to that seen in the younginid *Youngina*.

## Palate

The pterygoid is the largest palatal element preserved in this specimen and is a long flat bone that narrows to an anterior point (Fig 5). The right pterygoid is essentially complete anterior to the basicranial articulation, whereas only the central palatal portion of the left pterygoid is preserved. The ventral surface displays at least two prominent anteriorly extending fields of small teeth with a row of larger teeth positioned along the posterior edge of the transverse flange (Fig 5C). A single row of teeth are also present in the neodiapsid *Youngina* [10], and the

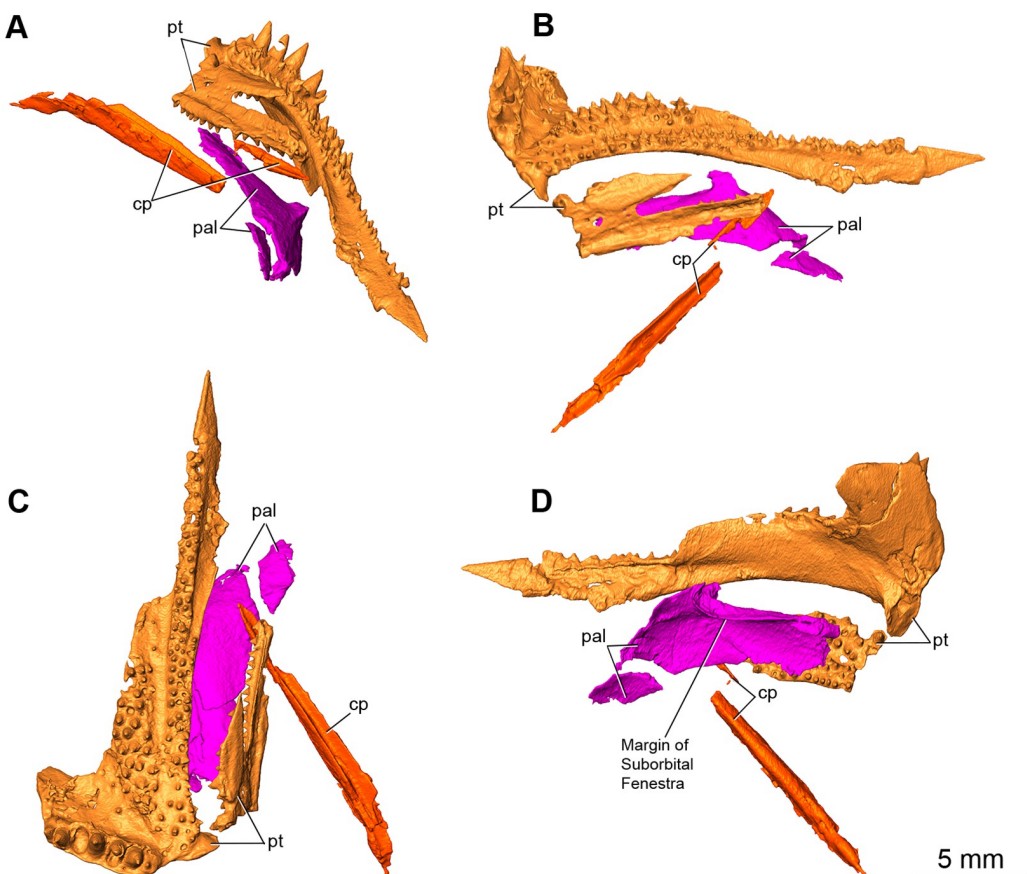

**Fig 5. Rendering of palatal elements present in *Maiothisavros dianeae* (ROMVP 87366) from CT image sequences.**
Palatal elements, including right and left pterygoid, palatine, and cultriform process in (A) anteroposterior, (B) left lateral, (C) ventral, and (D) dorsal views. Abbreviations: cp, cultriform process; pal, palatine; pt, pterygoid.

araeoscelid *Spinoaequalis* [9], but the teeth on the transverse flange of *Maiothisavros* are unusually large, similar in size to the dentary teeth. The lateral-most dental field extends anteriorly and is the widest row, displaying teeth of moderate size compared to those of the two other rows. The most medial dental field is the narrowest row, displaying the smallest teeth, extending the farthest anteriorly towards the snout. The enlarged teeth also exhibit slight recurvature sporting carinae, external plications, vertical reeding of the tooth apex in the posterior direction, and exhibit plicidentine at the base (Fig 2B), whereas some of the smaller teeth also exhibit some minor re-curvature. The overall degree of recurvature and mediolateral wide base of the larger palatal teeth is comparable to the teeth in the mid-region of the dentary, but they have a narrower apex pointed posteriorly. The transverse flange extends ventrally in a typical amniote fashion with a row of 7 large teeth running mediolaterally, with the largest teeth occupying the middle positions of the row (Figs 2B and 5C). Dental replacement is clearly present among the teeth of the transverse flange as one tooth position is empty, and a young, small tooth occupies another large socket on the transverse flange. Dental fields appear separated by an adjacent trough and are positioned on raised swellings of the pterygoid. The dorsal surface in the mid-region of the pterygoid is smooth and concave in shape, with a prominent medial groove extending anteriorly, whereas the posterior surface is flat and angled downward posteriorly (Fig 5D). The anterior portion of the right pterygoid displays an artificial rough texture due to the influence of hydrocarbon interference in the CT data.

Only the medial anterior portion of the left palatine is present (Fig 5). This element is a thin flat bone that may also include a non-distinct minimal portion of the vomer anteriorly, which does not have a discernable suture and as such is not segmented separately. The ventral surface of the palatine is smooth and concave with a pronounced ridge on the lateral side denoting the posterior margin of the internal naris (Fig 5D). Interestingly, there are no denticles present on the ventral surface, likely because this specific area of the palatine may not have any denticles, similar to that of *Orovenator mayorum*. The dorsal surface is smooth in texture and appears flat with a single pronounced protuberance on the medial side opposite to the posterior margin of the internal naris (Fig 5A and 5B). The lateral process is robust, and the size of the suborbital fenestra posterior to it is therefore wide and long, indicating the presence of a rather large suborbital fenestra, a hallmark of diapsids [8, 11].

A slender elongate bone located at the same level as the palatal elements is identified as the cultriform process of the parasphenoid. It is a trough shaped element that narrows anteriorly to a point (Fig 5). The anterior-most tip is broken off from the main body of the elongate bone, but it remains spatially associated with the main body of the cutriform process. The external surface lacks rugosity and appears smooth with no noticeable foramina, rugosities, or teeth.

## Mandible

The right ramus is relatively complete anteriorly with the component elements in closer association with each other than in the left ramus, but both lack most of the post dentary region (Fig 6).

The dentary is the largest mandibular element preserved in ROMVP 87366 (Fig 6). It is also a rather slender elongate element with a distinct flattened ventrolateral shelf that extends the length of the dentary until it merges with the ventrolateral shelf of the angular bone. One of the many distinctive features of this bone is the presence of an anterior expansion of the dentary at the level of the symphysis, forming a downward curving ventral process (Fig 6B). This ventral process is related to the expanded symphysis that conjoins the dentaries at the midline and is an unusual feature among Paleozoic tetrapods, especially when compared to the overall slender proportions of the bone. The slender lateral surface carries scattered foramina along its length below the level of the alveolar shelf, but surprisingly numerous small foramina are also concentrated on the anteroventral edge of the bone. Although *Azendohsaurus* and the aquatic *Tanystropheus* [32, 33] are separated phylogenically from each other, the concentration of foramina at the ventrally expanded symphyseal region of the dentary is somewhat similar to that seen in *Maiothisavros dianeae*. The slightly expanded posterior portion of the dentary has a V-shaped indentation on the flat lateral surface (clearly visible on the left dentary of ROMVP 87366 in Fig 6B) for the attachment of the surangular that appears to overlay the dentary and has a long anterior process running below the 9 most posterior tooth positions of the dentary. A prominent ventrolateral ridge gives the lateral surface of the dentary a distinctly concave appearance below the level of the alveolar shelf, in contrast to the more normal convex lateral surface that is commonly found in other parareptiles and eureptiles. The ventrolateral ridge separates the slightly concave outer surface of the dentary from a slightly convex but ventrally facing surface clearly visible in Fig 6A and 6B. The medial surface of the dentary has a prominent Meckelian groove extending along its middle in an anteroposterior direction and is narrowest at the ventral most portion of the dentary, but becomes anteroposteriorly wide along the entire length of the dentary.

It is important to stress the unique morphology of the dentary compared to the common amniote condition. The lateral surface of the dentary in *Maiothisavros dianeae* is flattened

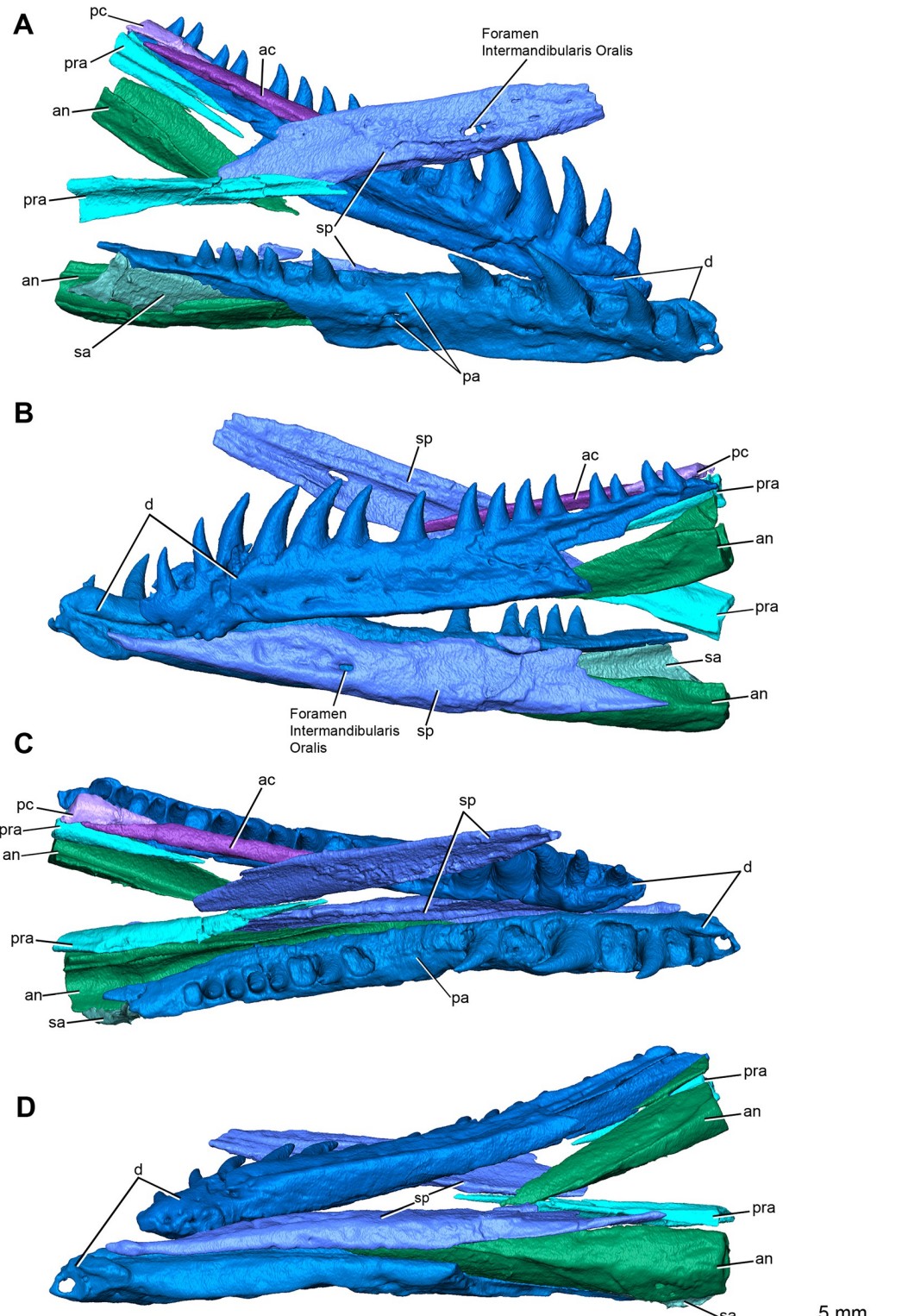

**Fig 6. Rendering of *Maiothisavros dianeae* (ROMVP 87366) mandible from CT imaging sequences.** Right and left dentary, right and left angular, right and left splenial, right surangular, anterior coronoid, and posterior left coronoids in (A) right lateral, (B) left lateral, (C) dorsal, and (D) ventral views. Abbreviation: an, angular; ac, anterior coronoid; d, dentary; pa, pathology; pc, posterior coronoid; pra, prearticular; sa, surangular; sp, splenial.

rather than rounded at the top and the occurrence of the ventrolateral ridge creates a concave lateral surface that extends posteriorly via the splenial and angular, rather than a typical convex surface found in other eureptiles or parareptiles [34]. Strikingly, the ventral surface of the dentary is flat, broad, and expanded medially and posteriorly by contributions of the splenial and angular to the ventral surface of the mandible; a condition converged upon by some later archosauromorphs, such as *Macrocnemus* [35]. This unusually flat and robust ventral surface readily distinguishes *Maiothisavros dianeae* from other Paleozoic tetrapods. Transverse sections of the dentary show that despite its apparent overall slenderness, this major element of the lower jaw is robust and thick beneath the tooth row (Fig 7).

Rather curiously, there is evidence of significant pathology on the right dentary of ROMVP 87366, extending through tooth positions 8 to 10 (Figs 6A and 7A–7C). This pathology has resulted in the formation of bone tissue over three tooth sockets, lateral rugosity visible on the lateral surface, and a prominent hole or foramina that connects the internal Meckelian canal with the outside (Fig 7B and 7C). Pathologies in Paleozoic amniotes are rare, and particularly rare for the Permo-Carboniferous with only one previous report on mandibular and dental pathology in an early Permian reptile [36].

Each dentary has places for 21 teeth and displays subthecodont dentition (Fig 7A and 7D). The teeth vary in height significantly, with the first three tooth positions (1 to 3) being smallest in size. There is an increase in tooth height and width moving posteriorly, with the largest teeth observed in positions 5 to 8, and a gradual decrease in size for the posteriormost teeth. The teeth are recurved and conical in shape, with plicidentine around the crown base and carinae on the mesial and distal edges of each tooth (Fig 2A). Exceptionally prominent carinae are only present on the largest teeth (5 to 8), curving laterally from the apex down the length of the tooth and are followed posteriorly by a prominent groove to form an effective piercing edge (Fig 2A). Interestingly, these conical teeth are tilted posteriorly and are mediolaterally wide at the base with a crown apex that is also pointed medially rather than being flattened, the commonly found condition in varanopid and sphenacodontid synapsids. The degree of recurvature for the smallest teeth located posteriorly is less than that of the larger more anterior teeth that display greater recurvature. The posterior tilting of the largest teeth is associated with a greater degree of curvature closer to the base of the tooth, whereas the smaller teeth curve posteriorly near the tooth apex.

ROMVP 87367, an additional fragmentary right dentary belonging to another individual of *Maiothisavros dianeae* was also segmented (Fig 8). Only the anterior-most portion of the right dentary is preserved and is smaller in size than the holotype, indicating that it was likely younger in age. This smaller specimen has 10 tooth positions, with teeth present in positions 5, 6, 8, and 10. The dentary bone and its dentition are indistinguishable from that in the larger holotype specimen, including the arrangement of lateral foramina and mandibular canal.

The splenial is the second-largest mandibular element, forming nearly the entire medial surface of the mandible (Fig 6B). The left splenial does not preserve the splenial symphysis and is missing more of the posterior portion compared to the right splenial. The right splenial is nearly complete and shows the anterior process that contributes to the mandibular symphysis. Unfortunately, the left splenial also has a rough medial surface, likely the result of hydrocarbon obstruction within the CT data. Both right and left splenial also clearly display the foramen intermandibularis oralis. Interestingly, the right splenial contributes to the symphysis as it expands medially and extends generously anteriorly to meet with the ventral process of the dentary. The splenial forms a long ventral suture with the anterior portion of the dentary and contributes to the nearly flat, distinct ventral surface for the anterior portion of the mandible (Fig 6D).

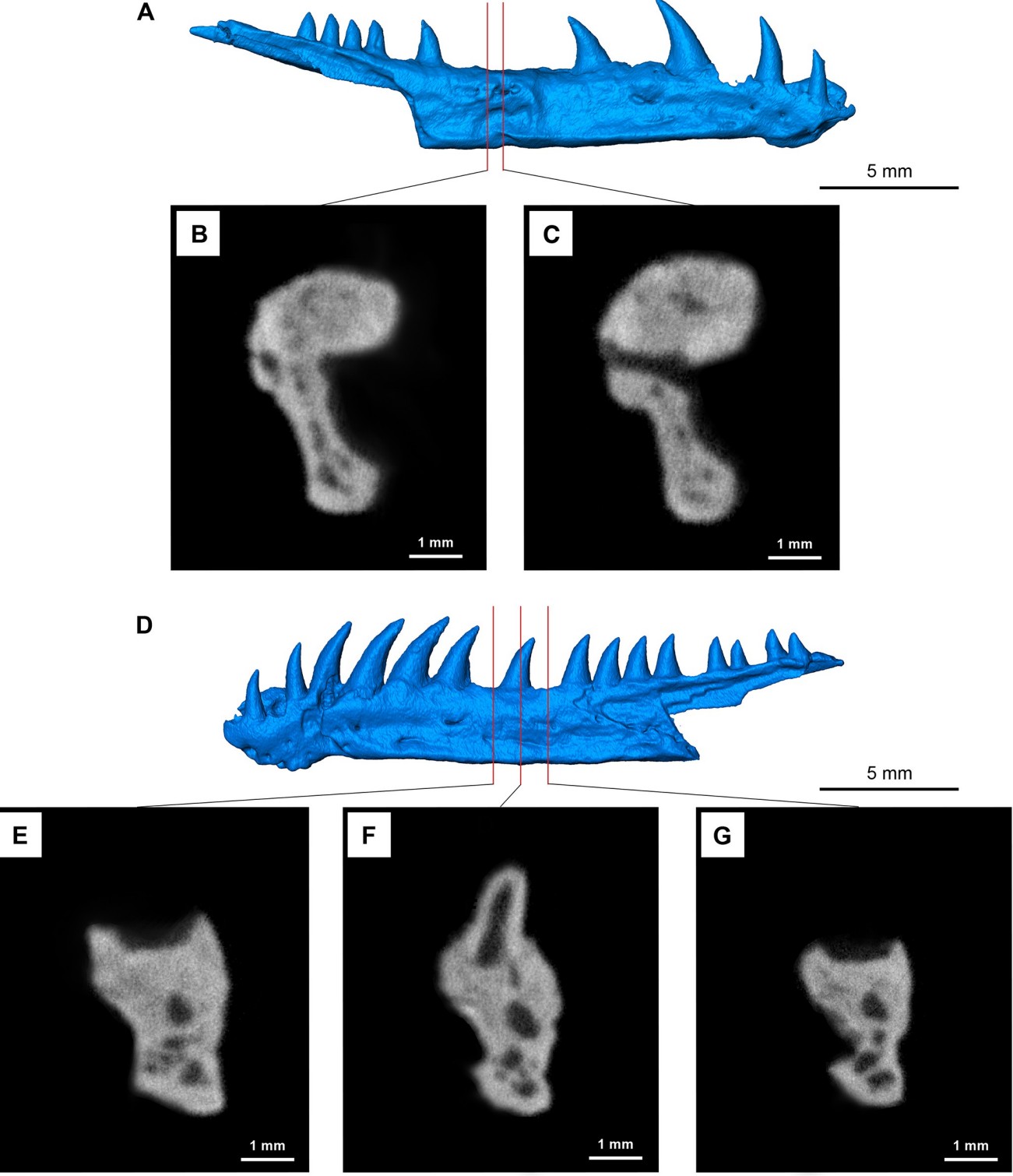

**Fig 7. Pathology of right dentary compared to the normal area in the left dentary for (ROMVP 87366).** (A) right and (D) left dentary renderings. Individual cross-sectional CT images for (B) and (C) displaying the pathology area, whereas (E), (F), and (G) display the same area but for the normal segment.

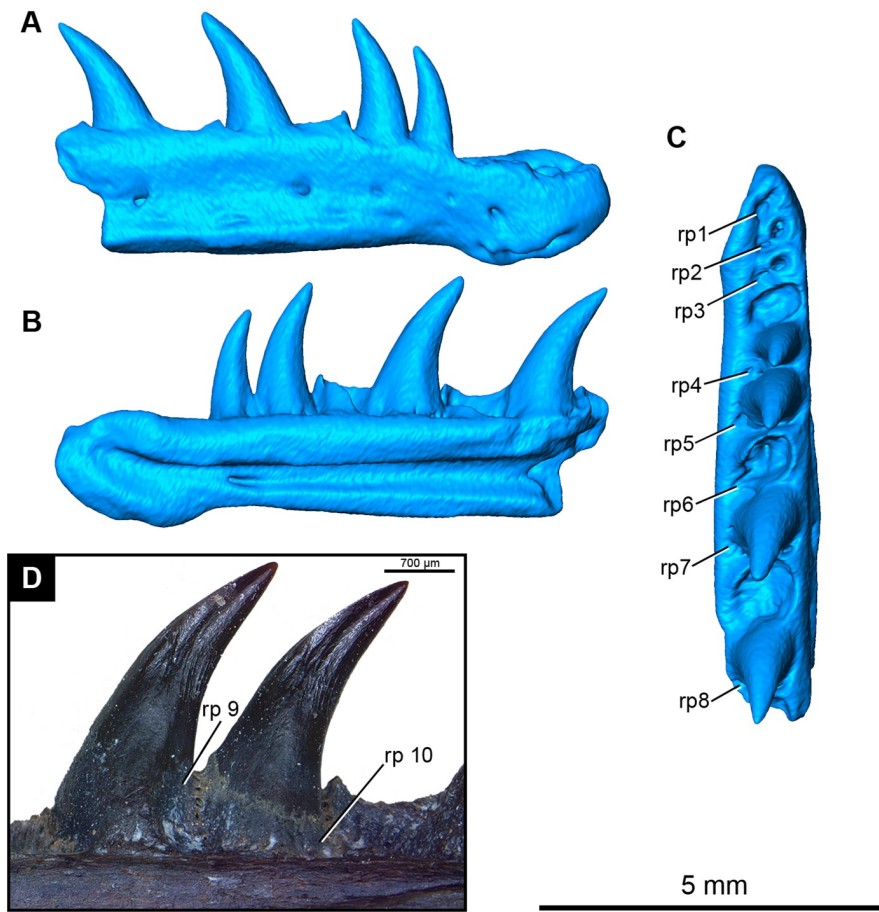

**Fig 8. Rendering and photograph of *Maiothisavros dianeae* right dentary.** ROMVP 87367, right dentary in (A) labial, (B) lingual, (C) occlusal, and (D) ROMVP 87368 photograph of tooth positions 8 and 9 in lingual view showing resorption pits. Abbreviation: rp, resorption pit.

Unfortunately, only the anterior portions of each angular are preserved (Fig 6). The right angular is more complete than the left angular but both lack most of the posterior region. It forms a suture with the posterior portion of the dentary and splenial, as well as the anterior ventral portion of the surangular. The preserved portion of the angular comprises close to the entirety of the posterior lateral and ventral surfaces of the mandible. Thus, the element has a robust U-shaped outline in cross-section at the level of the posterior dental region and has a slender anterior process contributing to the mandible between the splenial and dentary bones. Like the splenial and dentary bones, the angular has a considerably flat ventral surface that becomes more prominent posteriorly and makes a broad ventral contribution to the mandible, again a very unusual feature of this taxon. It also displays a modest ventrolateral ridge that joins with the ventrolateral ridge of the dentary to continue the robust ventral surface of the mandible posteriorly.

Only a limited portion of the right surangular is preserved and composes a small posterior lateral surface of the mandible (Fig 6). It is positioned superiorly to the angular and inferior to the posterior terminus of the dentary and appears as a slender, thin, slightly U-shaped element in cross-section. The surangular narrows anteriorly and would cover the corresponding slight depression on the lateral surface of the dentary just under the posterior terminus of the dentary.

The prearticular fragments are preserved medially on the posterior portion of the mandibles (Fig 6) and is a narrow slender element with a broad longitudinal groove. Neither is complete, and its original length is impossible to determine. The medial surface of the right prearticular prominently displays a noticeable ridge that narrows as it extends anteriorly. The lateral surface is largely composed of a wide groove that narrows anteriorly. The left prearticular terminates in an anterior bifurcation that only represents the most anterior portion of the prearticular.

There are two coronoid ossifications preserved on the left mandibular ramus of *Maiothisavros dianeae*. The anterior coronoid appears to be complete and is an extremely narrow, elongated, rod-shaped element positioned directly above the posterior region of the dentary on its medial side (Fig 6). The medial and dorsal surfaces of the anterior coronoid are completely smooth except for a dorsal groove extending posteriorly for attachment to the posterior coronoid. The anterior fragment of the left posterior coronoid is articulated dorsally to the posterior coronoid. The medial and lateral surfaces are completely smooth. While the anterior portion is slender and has a pointed terminus, the posterior region is expanded, likely forming the coronoid eminence of the process near the anterior end of the adductor fossa (Fig 6).

**Tooth attachment and implantation.** The histological section illustrates that the tooth root is fused directly to the jawbone with mineralization of the periodontal tissues and no periodontal space is visible; thus, the form of tooth attachment is ankylosis [37]. Additionally, the teeth sit on top of the dentary shelf and demonstrate attachment in an asymmetric shallow socket primarily on the labial side of the jawbone, illustrating subthecodonty as the form of tooth implantation (Fig 9B) [37, 38]. No jawbone is present overlaying the lingual surface of the teeth, rather the alveolar bone slightly extends over the base but does not reach above the labial surface of the jawbone. The dentine does not come into direct contact with the jawbone since the alveolar bone (highly vascularized attachment tissue [39]) is found in between. At the outer edge of the tooth surface, acellular cementum (mineralized tissue with no cell bodies present [37, 39, 40]) is around the tooth base on both the labial and lingual sides; however, the acellular cementum extends beyond the alveolar bone and jawbone only on the labial side (Fig 9B).

The tooth base of *Maiothisavros dianeae* has externally visible plications in the dentine that do not reach the tooth crown. The cross-section reveals that the plicidentine is situated at the base of the teeth, below the jawline (Fig 9D). The dentine consists of loose folds that do not contain primary or secondary branching; thus, the dentine does not form dentine lamellae in the center of each fold [41].

**Tooth development and replacement.** *Maiothisavros dianeae* appears to have an alternating pattern for tooth replacement, where every other tooth position contains a resorption pit or is missing its functional tooth, for two individual right dentaries, ROMV 87367 and ROMVP 87368 (Fig 8) [42]. A separate right dentary from another individual, ROMVP87369, displays two small replacement teeth in the process of erupting (Fig 10). The resorption pits that normally form in the center of the tooth base on the lingual surface, appear to be more posteriorly positioned, visible on the dentary bones (Fig 8).

The third tooth on the fragmentary left dentary of ROMVP 87370 exhibits a small resorption pit on the tooth base, with no replacement tooth in position (Fig 9A). The thin section reveals clear incremental lines of von Ebner within the dentine (Fig 9C). The functional tooth had a total of 221 incremental lines of von Ebner with a mean line width of 2.99 ± 0.51 μm, while the missing replacement tooth was estimated to have had approximately 29 lines. Thus, the estimated replacement rate for *Maiothisavros dianeae* was calculated to be approximately 192 days.

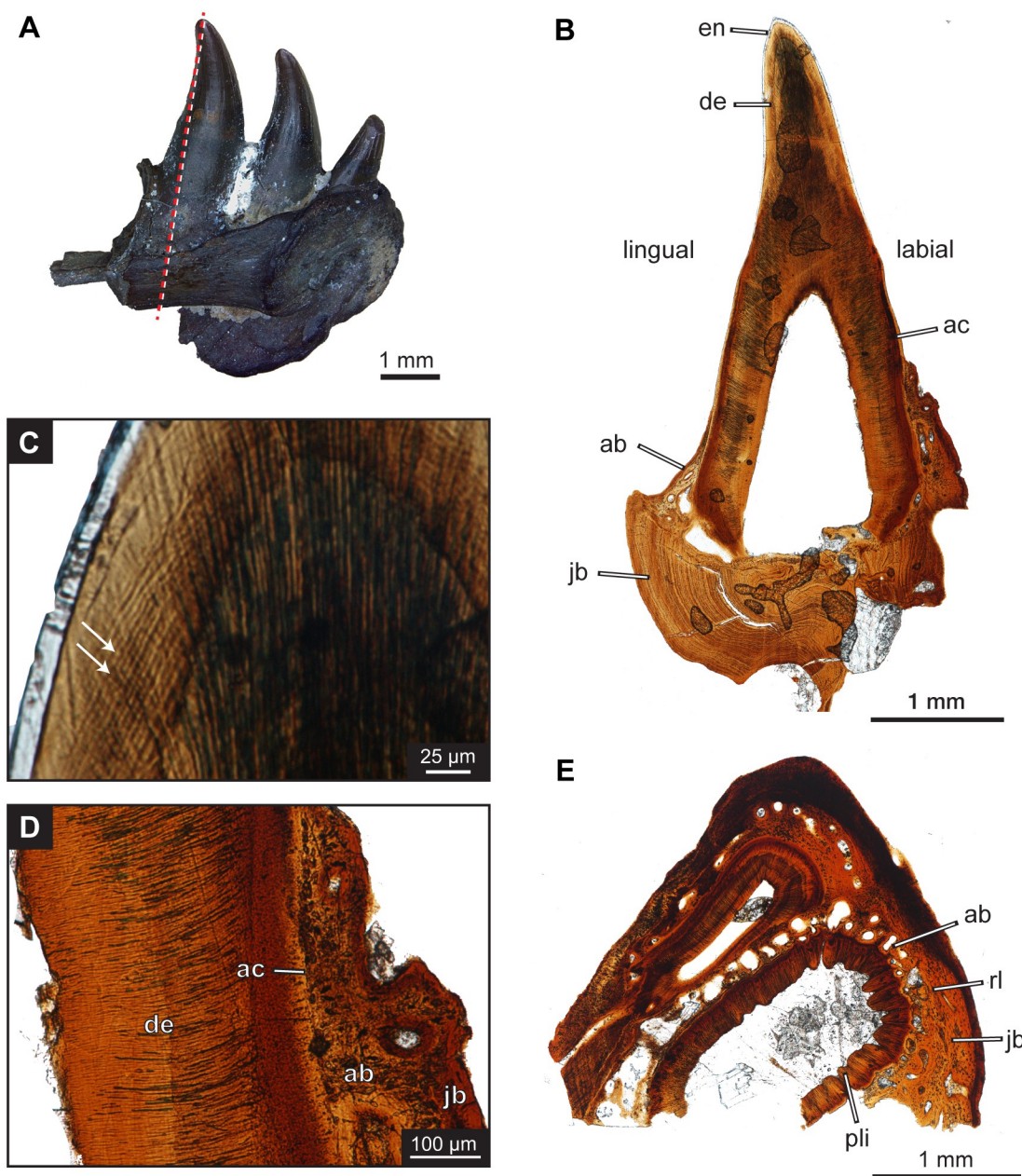

**Fig 9. Histology of *Maiothisavros dianeae* left dentary (ROMVP87370).** (A) Lingual view of fragmented left dentary with dashed red line showing the plane of the longitudinal section through the functional tooth. (B) Complete view longitudinal thin section at tooth position 3. (C) Close-up view of functional tooth section showing incremental lines, white arrows. (D) Close-up section showing tooth attachment tissues. (E) Cross-section near the tooth base showing plicidentine. Abbreviation: ab, alveolar bone; ac, acellular cementum; de, dentine; en, enamel; jb, jawbone; pli, plicidentine; rl, reversal line.

## Discussion

### Diagnostic morphologies

*Maiothisavros dianeae* is a highly unusual new small reptile from the early Permian and represents a significant new discovery, since the fragmentary remains of the skull indicate that it is a diapsid, possibly a neodiapsid. As such it would be only the second such reptile representing

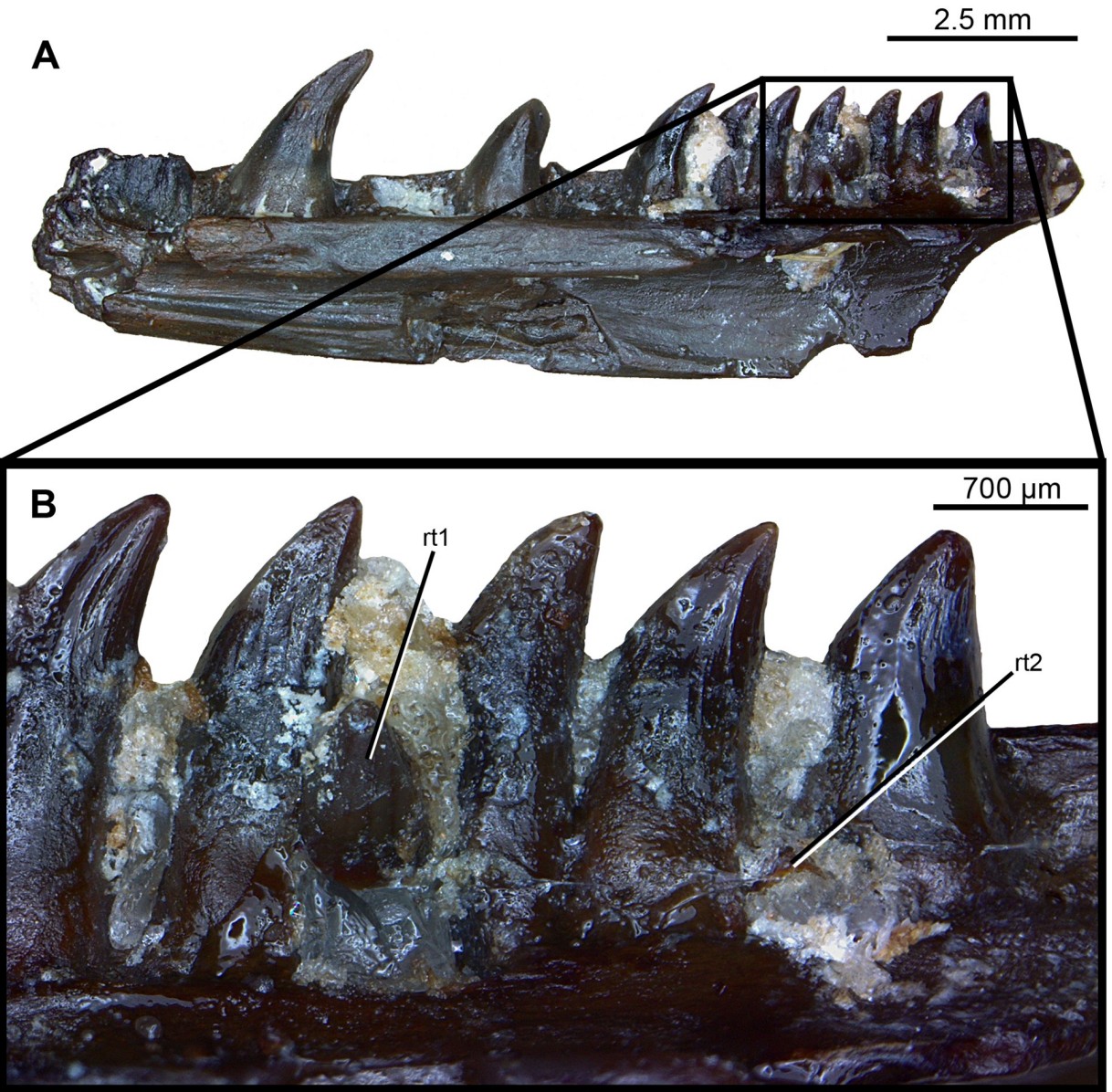

**Fig 10. Photograph of replacement teeth in positions for *Maiothisavros dianeae* (ROMVP 87369).** Right dentary in (A) lingual view and (B) close-up of two replacement teeth. Abbreviations: rt; replacement tooth.

the initial stages of diapsid evolution. *M. dianeae* is characterized by the presence of a distinctly slender lower jaw that possesses a ventrolateral ridge that not only gives the narrow lateral surface a slightly concave appearance, but also a distinct nearly flat ventral surface. The mandible as a whole is anteriorly narrow, and the dentary displays a striking ventral expansion to accommodate the symphysis. Importantly, the dentary differs from the general amniote condition in being ventrally robust in thickness, with additional contribution from the splenial, and angular bones. It is also worth noting that the anteroventral extension of the splenial has a slender contribution to the symphysis, reinforcing slightly the symphyseal surface of the dentary.

*Maiothisavros dianeae* also possesses strongly recurved marginal dentition on the premaxilla, maxilla, and dentary which are proportionately large and conical, not

mediolaterllary flattened teeth. In addition, the largest teeth are tilted posteriorly. The especially large teeth of the transverse flange of the pterygoid are also noteworthy, as they display plicidentine and are just as large and recurved as those of the posterior portion of the dentary and maxilla.

Functionally, the mediolaterally wide-based, recurved conical dentition combined with extensive dental fields of the pterygoid, implies that *Maiothisavros dianeae* was exceptionally well-adapted for a hunting strategy involving piercing and holding prey [30]. The enlarged symphysis of the splenial, ventrolateral shelf of the dentary with contributions from the jugal and splenial, and broad ventral surface of the mandible act to reinforce the structural integrity of the jaw under stress as a means of increasing bite force. The lack of serrations and occurrence of vertical reeding of the enamel surrounding the crown apex base of each tooth also suggest that the dentition is specialized for effectively piercing prey. The functional significance of these features suggests that *M. dianeae* was well-adapted for capturing and puncturing the hard exoskeleton of insects that were abundant during this time. It is also plausible that *M. dianeae* preyed upon smaller vertebrates that were also common at this locality.

## Supporting evidence for placement in diapsida

*Maiothisavros dianeae* is likely to be a diapsid based on the morphology of the jugal, maxillary, and palatine bones. Even though a recent study hypothesized that varanopids were diapsids [43], for the purpose of this study we subscribe to the hypothesis that varanopids are synapsids [44]. The jugal is clearly more derived than in araeosceloid diapsids, due to the presence of a gracile triradiate morphology with a slender subtemporal process and nearly flat, mediolaterally expanded dorsal process that separates the orbit and the lower temporal fenestra. The jugal is distinct from those found in varanopid synapsids in lacking any lateral ornamentation and reduced sutural contact with the quadratojugal. Although the latter bone is not preserved, the jugal morphology reveals the sutural morphology with it. The maxillary and the palatine bones, although incomplete, are sufficiently well preserved to show the presence of a large sub-orbital fenestra, a synapomorphy of diapsid reptiles. The morphology of the main maxillary canal recovered from the neutron CT data is also characteristically diapsid [45].

The presence of only a few lateral foraminal projections from the main maxillary canal of *Maiothisavros dianeae* is similar to those found in the early Triassic archosauromorph diapsid *Prolacerta broomi*, which also exhibits a more channelized maxillary canal with fewer large lateral foramina [45]. In fact, both *Prolacerta broomi* and *Orovenator mayorum* possess a simple tubular canal shape of the main maxillary canal that is likely characteristic of the diapsid condition [45] and distinct from those in varanopid synapsids. Interestingly, the morphology of the anterior medial foramen for the superior alveolar nerve is rather thick in *Prolacerta broomi* and *Orovenator mayorum* for the main maxillary canal directly posterior to the dorsal projection of the maxillary canal, which contrasts with the rather narrow condition seen in *Maiothisavros dianeae*. However, the overall similar morphology of the maxillary innervation between *Maiothisavros dianeae* and *Prolacerta broomi* may also be interpreted as a possible placement of this new genus closer to archosauromorphs.

The simple tubular shape of the main maxillary canal exhibited by *Maiothisavros dianeae* also is in stark contrast to the condition seen in synapsids such as *Heleosaurus scholtzi* and *Varanosaurus acutirostris* [45], as the canals in these synapsids have a heavily branched, nontubular shape of the main maxillary canal supporting numerous lateral projections meandering in several directions.

## Comparison with *Orovenator mayorum*

*Maiothisavros dianeae* is unique among the other diapsids currently known from the Richards Spur locality, including *Orovenator mayorum* and possibly an unnamed diapsid [1, 4]. It is worth noting that a direct comparison between *M. dianeae* and the unnamed diapsid described by Carroll (1968) is not possible since the latter is based solely on an isolated parietal and the *M. dianeae* holotype does not preserve the parietal bone.

The jugal of *Maiothisavros dianeae* is gracile and characteristically triradiate in shape. It also importantly lacks the varanopid synapsid condition of lateral suborbital rugosities, which coincides with the diapsid condition found in *Orovenator mayorum* [4, 5]. There are several morphological differences between the jugal of *M. dianeae* and *O. mayorum*. The dorsal process of *M. dianeae* is angled more vertically than that of *O. mayorum*, which is oriented noticeably more in the posterior direction. The lateral surface seen in *M. dianeae* is also anteroposteriorly narrow, contrasting the broader condition seen in *O. mayorum*. Interestingly, the overlapping sutural surface between the dorsal process of the jugal and the postorbital in *M. dianeae* is more anteriorly oriented and does not extend as far ventrally compared to the more laterally oriented and downward extending condition observed in *O. mayorum*. The dorsal process of *M. dianeae* is discernably longer than the posterior process, which contrasts the condition of *O. mayorum*. The posterior process of *O. mayorum*, is also very delicate in contrast to the more robust bifurcate condition of *M. dianeae*.

The dentition of the dentary exhibited by *Maiothisavros dianeae* is larger, much less numerous, and displays a greater degree of recurvature than those of *Orovenator mayorum*. *M. dianeae* also displays a greater range of shape and size based heterodonty. The largest teeth are seen in positions 5 to 8 displaying the greatest recurvature and the most prominent vertical reeding of the plicidentine, which gradually becomes less prominent as the teeth decrease in size near the posterior terminus of the dentary. In contrast, *O. mayorum* has isodont dentition (similar size and shape) and the teeth only decrease in size after tooth position 26 [5].

The ventrolateral ridge extending the length of the dentary seen in *Maiothisavros dianeae* contrasts the slender anteriorly narrow shape of the dentary found in *Orovenator mayorum*, but both display a concentration of foramina in the anterior most portion of the dentary [5]. The ventrolateral shelf of the dentary also contributes a greater surface area to the ventral portion of the mandible than that of *O. mayorum*. However, perhaps most strikingly *M. dianeae* possesses the unique ventral expansion of the anterior tip of the dentary to accommodate the symphysis.

## Histology

The plicidentine observed in *Maiothisavros dianeae* consisted of loosely folded dentine, only present at the base of the tooth and not extending towards the crown. This condition is similar to *Delorhynchus* which also exhibited loosely folded dentine at the base [46], which also is unlike the highly folded dentine lamellae found for the large premaxillary teeth of *Colobomycter pholeter*. It has been previously noted that plicidentine may serve a structural support role as it anchors the base of the tooth to the jawbone [41, 46], and this may be important for *M. dianeae* since the teeth exhibit subthecodont implantation. Additionally, the alternating replacement pattern observed for *M. dianeae* has been suggested to be the classic primitive condition for many tetrapods [42]. The positioning of the resorption pits is peculiar since they normally develop in the medial region of the tooth base, whereas in *M. dianeae* the pits have shifted position to be more posterior. This migration has been observed in both large and small specimens, indicating that it is not an ontogenetic characteristic. This characteristic has also been noted for *Mosasaurus*, with the replacement teeth forming in the posterior-lingual

position [38]. Lastly, *M. dianeae* appears to have a long replacement period (192 days) and long-lived teeth (221 days), compared to other insectivorous species that have small, delicate teeth [47].

## Conclusion

*Maiothisavors dianeae* is an exceptional early Permian diapsid, unlike any previously described from the Permo-Carboniferous. The morphology of the triradiate jugal lacking lateral ornamentation, only discovered through neutron tomography, the anatomy of the palatine bone indicating the presence of a large suborbital fenestra, and the morphology of the main maxillary canal that was also recovered from the neutron CT data are characteristic of diapsids [45]. Clearly, the unique morphology of the mandible suggests a specialized predatory ecological role for this animal found within the upland Paleozoic faunal assemblage preserved at the Richards Spur locality. The proportionately large, recurved, mediolaterally wide, posteriorly tilted, conical dentition sporting medially trending carinae on the mesial and distal cutting edges (coronated vertical striations) and vertical reeding on the crown are indicative of a hunting strategy adept for piercing and holding prey. Interestingly, the mandible and most notably the dentary are rather slender, giving *M. dianeae* a very gracile appearance. However, the large, specialized piercing dentition indicate a higher bite force, likely facilitated by the unprecedented mandibular morphology not seen in any previously described Paleozoic amniote. It is also peculiar that the resorption pits where the replacement process starts on the dentigerous elements are located more posteriorly and that the longer than anticipated tooth replacement rate of 190 days occurs for such delicate teeth.

In conclusion, although clearly a highly autapomorphic diapsid, the available material is insufficiently informative to allow us to undertake a phylogenetic analysis to evaluate its position within Diapsida and in particular among Neodiapsida. We suggest that despite their fragmentary nature, the known remains show a surprisingly large number of autapomorphies, and its actual position within Neodiapsida may be only determined with new additional discoveries of more complete materials.

## Acknowledgments

We would like to thank Bill and Julie May for retrieving these materials from Richards Spur, Oklahoma and allowing us to study them. EDM would also like to thank Dylan Rowe and Kayla Bazzana for CT segmentation assistance, and Diane Scott for fossil preparation and photography. We thank the external reviewers David Ford, Sean Modesto, and Adam Pritchard for their thoughtful and very helpful comments and suggestions that improved significantly the manuscript.

## Author Contributions

**Conceptualization:** Robert R. Reisz.

**Data curation:** Joseph J. Bevitt, Robert R. Reisz.

**Formal analysis:** Ethan D. Mooney, Tea Maho.

**Methodology:** Joseph J. Bevitt.

**Project administration:** Robert R. Reisz.

**Visualization:** Ethan D. Mooney, Tea Maho.

**Writing – original draft:** Ethan D. Mooney.

**Writing – review & editing:** Ethan D. Mooney, Tea Maho, Joseph J. Bevitt, Robert R. Reisz.

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
