## [Decision Letter · Decision Letter 0]

29 Jul 2022

PONE-D-22-17394An intriguing new diapsid reptile with evidence of mandibulo-dental pathology from the early Permian of Oklahoma revealed by neutron tomographyPLOS ONE

Dear Dr. Reisz,

Thank you for submitting your manuscript to PLOS ONE. After careful consideration, we feel that it has merit but does not fully meet PLOS ONE’s publication criteria as it currently stands. Therefore, we invite you to submit a revised version of the manuscript that addresses the points raised during the review process. Three reviewers gave comments from different angles, please reply on their concerns , especially on the tooth morphology. For the phylogenetic position, you may have tried to code it in the previous matrice, but may not get a good result. If so, it is not necessary to include it as suggested by one reviewer.

We look forward to receiving your revised manuscript.

Kind regards,

Jun Liu

Academic Editor

PLOS ONE

Journal Requirements:

2. In your manuscript, please provide additional information regarding the specimens used in your study. Ensure that you have reported specimen numbers and complete repository information, including museum name and geographic location.

'No permits were required for the described study, which complied with all relevant regulations.

"This study was supported by a Natural Sciences and Engineering Research Council (NSERC) scholarship to T.M., and NSERC grant to R.R.R., and the Jilin University, China."

"Supported by a Natural Sciences and Engineering Research Council (NSERC) scholarship to T.M., and NSERC grant to R.R.R., and the Jilin University, China."

5. Please take this opportunity to be sure you have met all of our guidelines for new species. For proper registration of a new zoological taxon, we require two specific statements to be included in your manuscript.

a. In the Results section, the globally unique identifier (GUID), currently in the form of a Life Science Identifier (LSID), should be listed under the new species name, for example:

Anochetus boltoni Fisher sp. nov. urn:lsid:zoobank.org:act:B6C072CF-1CA6-40C7-8396-534E91EF7FBB

Another LSID for the manuscript itself should also appear within the Nomenclature statement. You will need to contact Zoobank (zoobank.org/About) to obtain a GUID (LSID). You should receive one LSID for your manuscript and a separate, unique LSID for the new species. 

b. Please also insert the following text into the Methods section, in a sub-section to be called ""Nomenclatural Acts"":

The electronic edition of this article conforms to the requirements of the amended International Code of Zoological Nomenclature, and hence the new names contained herein are available under that Code from the electronic edition of this article. This published work and the nomenclatural acts it contains have been registered in ZooBank, the online registration system for the ICZN. The ZooBank LSIDs (Life Science Identifiers) can be resolved and the associated information viewed through any standard web browser by appending the LSID to the prefix """" ext-link-type="uri" xlink:type="simple">http://zoobank.org/"". The LSID for this publication is: urn:lsid:zoobank.org:pub: XXXXXXX. The electronic edition of this work was published in a journal with an ISSN, and has been archived and is available from the following digital repositories: PubMed Central, LOCKSS [author to insert any additional repositories].

All PLOS ONE articles are deposited in PubMed Central and LOCKSS. If your institute, or those of your co-authors, has its own repository, we recommend that you also deposit the published online article there and include the name in your article.

Following a recent ruling by the International Commission on Zoological Nomenclature, electronic journals are now a valid format for publication of new zoological taxa. In order to ensure the valid publication of your new species, please be sure to include the updated version of Nomenclatural Acts (above). A complete explanation of our guidelines for publishing new species can be found on our website: http://www.plosone.org/static/guidelines#zoological.

Reviewers' comments:

Reviewer's Responses to Questions

**Comments to the Author**

1. Is the manuscript technically sound, and do the data support the conclusions?

Reviewer #1: Yes

Reviewer #2: Yes

Reviewer #3: Yes

2. Has the statistical analysis been performed appropriately and rigorously? 

Reviewer #1: N/A

Reviewer #2: N/A

Reviewer #3: N/A

3. Have the authors made all data underlying the findings in their manuscript fully available?

Reviewer #1: Yes

Reviewer #2: Yes

Reviewer #3: Yes

4. Is the manuscript presented in an intelligible fashion and written in standard English?

Reviewer #1: Yes

Reviewer #2: Yes

Reviewer #3: Yes

5. Review Comments to the Author

Reviewer #1: PLOS ONE Review

Manuscript Number: PONE-D-22-17394

Title: An intriguing new diapsid reptile with evidence of mandibulo-dental pathology from the early Permian of Oklahoma revealed by neutron tomography

The authors have presented a detailed description of new material from the Richards Spur locale consisting principally of mandibular and palatal elements, but also some cranial material, including an incomplete jugal and maxilla. They conclude that the holotype and 4 referred specimens are attributable to a new genus and species of diapsid reptile (Maiothisavros diane), based on the likely presence of a lower temporal fenestra, a suborbital opening and the morphology of the maxillary canal. They also recognise some pathology of the right dentary in one referred specimen. They further conclude that histological analysis of the marginal dentition suggests ‘pleurodont’ tooth implantation and a relatively slow rate of tooth replacement.

As the authors suggest, diapsid material from the Late Carboniferous and Lower Permian is very rare and any addition to our knowledge of early reptile diversity and evolution is welcome. For this reason, I fully support the publication of this article. However, there are several revisions and amendments that I propose the authors consider prior to publication, which are detailed below.

Abstract (lines 25-26). “Notwithstanding the hypothesis that varanopids are diapsids rather than synapsids…”. The authors are right to mention this alternative hypothesis, but I note it is restricted to the abstract only and omitted from the main text. I think it appropriate to include this in the main text, especially since the text implicitly follows the ‘varanopids as synapsids’ hypothesis.

Abstract (Lines 42-43) “…pleurodont implanted dentition reminiscent of extant snakes…” – please refer to comments in ‘Tooth attachment and Implantation (lines 490 – 492)’ below.

Introduction (line 51) “…now dated 316M.” The age of the Joggins material has certainly been subject to some debate. Benton et al. (2015) have suggested 318Ma, based on comparisons with the Langsettian European time unit. However, authors may have a more recent assessment, in which case a reference would be appropriate. Also ‘316 M’ should read ‘316 Ma’.

Introduction (line 52) “…Protoclepydrops…” – should read “Protoclepsydrops”. Also, there is some ongoing discussion on the affinities of the material assigned to this taxon - see Mann and Patterson (2020).

Introduction (lines 52-54, sentence ending “…in the fossil record.” The sentence doesn’t read well. Perhaps substituting ‘however’ for ‘but’ in line 54?

Introduction (lines 57-58, sentence ending “…turtles and squamates.” This implies that both turtles and squamates only began to diversify after the end Cretaceous extinction, whereas certainly squamates had already undergone significant diversification by the end of the Cretaceous (see Ricklefs et al. 2007).

Introduction (line 60,) sentence commencing Although amniotes synapsids….”. I suggest a small revision – perhaps “Although amniotes, in particular synapsids, parareptiles etc…”.

Histology (lines 141 – 142). Repetition of “…prior to histological analysis.”.

Locality (line 191). Th date range 289-286 Ma should be described as Artinskian age (see International Chronostratigraphic Chart v.2022/02).

Diagnosis (Lines 198-199) “…. differs from varanopids in lacking any lateral suborbital ornamentation.” Whilst this statement correctly reflects the condition found in derived varanopids, jugal ornamentation does not occur in the ‘basal’ varanopids Archaeovenator (Reisz and Dilkes 2003) and Ascendonanus (Spindler et al. 2018), both of which possess slender triradiate jugals with narrow suborbital and subtemporal rami.

Skull Roof (line 227) “…2 in-situ teeth”. It is difficult to determine from figure 1B that these two teeth can be confidently described as ‘in-situ’, since no trace of the underlying premaxilla remains. Therefore, the assessment of the number of teeth in the premaxilla is likely to be somewhat speculative.

Skull Roof (line 242). Delete “...a...”.

Skull Roof (line 297) “The subtemporal or posterior process…”. This ramus of the jugal has already been introduced in line 293. So perhaps delete “...or posterior…”?

Skull Roof (line 299). The jugal in the varanopids Archaeovenator and Ascendonanus are very similar in shape to the jugal in ROMVP 87366. So, it is a little misleading to imply all varanopids possess a more robust process.

Skull Roof (lines 300 – 301) “…lacks the kind of rugosity seen in varanopids…”. Refer to note under ‘Diagnosis’ above. Perhaps amend to “some varanopids”.

Fig.5. It is difficult to distinguish between the cultiform process and the pterygoid, certainly in image A. Perhaps a more distinctive colour can be used for the cultriform process?

Palate (line 315). It appears from Fig.5 that these might be better described as ‘fields’ rather than ‘rows’, since the teeth are arranged randomly within each distinctive field. This logic is applied in line 317. The teeth on the posterior margin of the transverse process are indeed a ‘row’ of single teeth.

Palate (lines 335 – 336). It would be helpful to show the possible ectopterygoid in a different colour scheme or at the very least identify it in the figure.

Palate (lines 339 – 341) sentence beginning “The ventral surface…”. It would be useful for a call-out to indicate which of the 4 images of Fig.6 reflect the elements described in the text and perhaps a label as well (i.e., the margin of the internal naris, the lateral process indicating the maxillary ramus of the palatine etc.). The latter is important since the presence of a suborbital opening (an important diagnostic feature in determining the diapsid affinities of this taxon) should be evidenced as clearly as possible.

Mandible – general. Again, the text lacks specific call-outs for each of the 4 images of Fig.6 with respect to important and distinctive morphological observations (e.g., the anteroventral expansion of the dentary).

Mandible (lines 388 and 404). Suggest this should read “…other eureptiles and parareptiles.” Since it currently implies the taxon is a parareptile. Since this is also an unusual morphology among synapsids, perhaps “…other early amniotes.” may fit better.

Mandible (line 417) “…. displays pleurodont dentition”. Refer to comments in ‘Tooth attachment and Implantation (lines 490 – 492)’ below.

Mandible, paragraph commencing “Each dentary has…”. In general, there should be more call-outs to figures, particularly when discussing tooth positions.

Mandible, paragraph commencing “The splenial is…”. Please provide more specific call-outs to individual images in Fig. 6 when describing anatomical detail.

Mandible, line 447. Do the authors mean to say “splenial symphysis” or would “mandibular symphysis” be more accurate?

Mandible (line 450). Should this read “the internal or lateral surface contains the same large foramen as the medial surface….”?

Tooth attachment and Implantation (lines 490 – 492). I agree that tooth attachment appears to be ankylosed to the dentary via the mineralization of periodontal tissue. In addition, the dentary is higher labially than lingually. However, this does not necessarily indicate pleurodonty sensu stricto. In Bertin et al. (2018) – the reference applied in the text – ankylosis to the dentary via mineralized attachment tissue, with no periodontal space, is associated with subthecodont implantation (see Bertin et al. Fig. 2B). Bertin et al. define subthecodonty as implantation in an asymmetrical shallow socket, (Fig.3E), rather than a pleurodont implantation where there is a distinct gradient between the labial and lingual margins (Fig. 3C). Text Fig. 9B is more reflective of a subthecodont condition as defined by Bertin et al., (although I have heard such implantation described as subthecodont and subpleurodont). In any event, the similarity of this implantation to snakes (as suggested in the abstract) is questionable.

Diagnostic Morphologies (lines 547 – 548). The flattened ventral surface had already been noted in lines 544 – 545, so this seems a little repetitive.

Diagnostic Morphologies. It is interesting to note that the authors have omitted to mention here the presence of both an anterior and posterior coronoid in Maiothisavros diane. This is an important feature of early amniote phylogeny. Among the majority of early diapsids, indeed early sauropsids in general, there is only a single coronoid. Two coronoids have only been reported in Hylonomus (Carroll 1964, although somewhat contentious) and Araeoscelis (Vaughn 1955). Indeed, the similarity with the latter taxon might suggest an alternative hypothesis to the possible neodiapsid affinities of M. diane. So, the clear identification of two coronoids in Maiothisavros diane is a key anatomical observation. Two, or even three, coronoids are present in stem-amniotes. Generally, two coronoids are present in synapsids. Only one is present in the varanopids Archaeovenator and Mesenosaurus. The implications of two coronoids in M. diane is therefore certainly worthy of reference.

Supporting evidence for placement in Diapsida (lines 572 – 573) “…varanopid synapsids, in lacking any lateral ornamentation….”. Please refer to comments in ‘Diagnosis (Lines 198-199)’ above.

Supporting evidence for placement in Diapsida (lines 576 – 577) “…hallmark feature...”. If the authors consider this a synapomorphy of diapsid reptiles, perhaps this would be a more appropriate wording.

Supporting evidence for placement in Diapsida (lines 576 – 577), sentence commencing “Interestingly, the morphology….”. This sentence is a little confusing. The morphology of the anterior medial foramen is not the issue, but rather the shape of the maxillary canal directly ventral to it.

Supporting evidence for placement in Diapsida (lines 588 – 590), sentence commencing “However, the similar morphology….”. It is not clear what the authors are suggesting here. If they infer that there may be a closer phylogenetic affiliation between M. diane and archosauromorphs, I should advise caution. At least until it can be tested for congruence within a phylogenetic framework.

Comparison with Orovenator mayorum (line 604) “…varanopid condition of lateral suborbital rugosities…”. Please refer to comments in ‘Diagnosis (Lines 198-199)’ above.

Comparison with Orovenator mayorum (line 609) “Interestingly…”. Perhaps a little too repetitive given the same adjective is used in line 606.

Comparison with Orovenator mayorum (line 614) “The sub orbital process is underlain by the maxilla in O. mayorum…”. The authors do not follow this statement with a comparison for M. diane, so it is somewhat redundant. Also, “O. mayorum” is not italicized.

Histology (line 639) “…pleurodont dentition…”. Please refer to notes under ‘Tooth attachment and Implantation (lines 490 – 492)’ above.

Histology (line 641). “…it…” should read “...is...” and “…normal…” should read “…normally…”

Conclusion (lines 699 – 671), partial sentence commencing “We propose that…” and ending “…in younginiforms.” I cannot agree that the number of autapomorphies in any given taxon provides an indication of a more crown-ward phylogenetic position. Perhaps this phrase is used by the authors in respect to lines 588 -590 (see above). In which case I would continue to advise caution.

Conclusion (lines 673 – 675) sentence commencing “In conclusion…”. This sentence basically repeats the sentence commencing on line 667 in the concluding paragraph.

Dr. David Ford

5/7/22

References:

Benton, Michael J., Donoghue, Philip C.J., Asher, Robert J., Friedman, Matt, Near, Thomas J., and Vinther, Jakob. 2015. Constraints on the timescale of animal evolutionary history. Palaeontologia Electronica 18.1.1FC; 1-106. https://doi.org/10.26879/424

Bertin, T.J., Thivichon-Prince, B., LeBlanc, A.R., Caldwell, M.W. and Viriot, L., 2018. Current perspectives on tooth implantation, attachment, and replacement in Amniota. Frontiers in physiology, 9, p.1630.

Carroll, R.L., 1964. The earliest reptiles. Zoological Journal of the Linnean Society, 45(304), pp.61-83.

Mann, A., and Paterson, R.S., 2020. Cranial osteology and systematics of the enigmatic early ‘sail-backed’ synapsid Echinerpeton intermedium Reisz, 1972, and a review of the earliest ‘pelycosaurs’. Journal of Systematic Palaeontology, 18(6), pp.529-539.

Ricklefs, R.E., Losos, J.B. and Townsend, T.M., 2007. Evolutionary diversification of clades of squamate reptiles. Journal of evolutionary biology, 20(5), pp.1751-1762.

Reisz, R.R. and Dilkes, D.W., 2003. Archaeovenator hamiltonensis, a new varanopid (Synapsida: Eupelycosauria) from the Upper Carboniferous of Kansas. Canadian Journal of Earth Sciences, 40(4), pp.667-678.

Spindler, F., Werneburg, R., Schneider, J.W., Luthardt, L., Annacker, V. and Rößler, R., 2018. First arboreal' pelycosaurs'(Synapsida: Varanopidae) from the early Permian Chemnitz Fossil Lagerstätte, SE Germany, with a review of varanopid phylogeny. PalZ, 92(2), pp.315-364.

Vaughn, P. P. 1955. The Permian reptile Araeoscelis re-studied. Harvard Museum of Comparative Zoology, Bulletin 113:305-467.

Reviewer #2: The authors describe a partial skull and mandible from the lower Permian Richards Spur locality of Oklahoma, U.S.A., as a new genus and species of diapsid reptile. I have no problem with the identification of the specimen as a diapsid reptile, and I do agree that the material is too limited to justify a new phylogenetic analysis, nor do I expect the authors to resolve the question of varanopids-as-diapsids at this time on the basis of such paltry material. However, I do not think that the new diapsid is as unique as the authors claim. For instance, the authors interpret the tooth implantation as pleurodont, but that interpretation is at odds with recent histological work on pleurodont squamate teeth. The authors also describe the dention as “snake-like” but the only teeth that look snake-like to me are the maxillary caniniforms and those forming the caniniform region in the dentary (it is hard to tell for the premaxillary teeth); the remaining teeth are stout with recurved tips but are not remarkably snake-like.

MAJOR COMMENTS

The authors suggest that the new diapsid may be a neodiapsid, but the only character that supports a neodiapsid affinity is the slender subtemporal process of jugal. The maxillae exhibit caniniform teeth (although the authors refer to these teeth only as “larger teeth”—however the pulp cavities of these teeth look to be the same size as the teeth immediately preceeding and succeeding the caniniform teeth!), unlike Paleozoic neodiapsids, and the lower temporal fenestra is closed (the jugal exhibits sutural surface for the quadratojugal).

I disagree that tooth implantation is pleurodont. The histological sections provided by the authors indicate that the labial and lingual dentine walls are the same length and thus symmetrical. This is unlike the pleurodont teeth described by LeBlanc et al. (2020) for iguanids. In figure 9b the tooth sits in a shallow socket and implantation appears to be subthecodont; i.e. the jaw bone exhibits no pleura.

General: where authors state bone X forms a suture with bone Y, I think it would be appropriate to identify the kind of suture, e.g. “the angular forms overlapping sutures with the dentary and the splenial” or “the jugal appears to have formed a scarf joint with the quadratojugal”.

OTHER COMMENTS:

Line 36: change “while” (connotes time) to “whereas” (comparative). See also lines 241, 309, 332, 336, and 642.

Line 55: change “Diapsids represents” to “Diapsids represent”

Line 60: change “Although amniotes synapsids, parareptiles, and non-diapsid eureptiles” to “Although synapsids, parareptiles, and non-diapsid eureptiles”

Lines 42-45: change these lines to “Lake Shale of Kansas [3], already was characterized by the presence of a suborbital fenestra, bound by the palatine, ectopterygoid, and maxilla [4], and two temporal fenestrae, consisting of a lower or inferior temporal fenestra bound by the jugal, quadratojugal, postorbital, and squamosal, and an upper or superior temporal fenestra bound by the postorbital, parietal, and squamosal.” [note addition of commas]

Line 76: change “Early Permian” to “early Permian”

Line 87: change “Reisz, et al (2011a)” to “Reisz et al. [5]”

Line 115: is the matrix of the specimen limestone? I thought it would be clay or marlstone.

Line 171: I think the new species name should follow line 170 here as “Maiothisavros diane gen. et sp. nov.” otherwise the following section (Etymology) makes no sense.

Line 177: if you are honouring Diane with a specific epithet then it should be “dianeae” (-ae suffix). If you intend to keep the specific epithet as “diane” you should declare it a noun in apposition.

Line 181: change “mandibles” to “mandible” [note that each gnathostome has a single mandible that can be subdivided into left and right mandibular rami]

Lines 194-199: the presence of an anterior coronoid is rare among Paleozoic reptiles in general and in diapsids in particular, and should be included in the Diagnosis

Line 203: change “mandibles” to “mandible”

Line 214: with regards to “mediolateral thickness of the existing dentition” do you mean “labiolingual thickness of the existing dentition” ?

Line 215: change “a capture and kill predatory behavior” to “a capture-and-kill predatory behavior”

Line 230: with regards to “mediolateral” here I think you should be using dental terminology such as labial, ligual, distal, and mesial for orientation. See also line 243.

Line 233: “width” is not a helpful term when describing teeth: do you mean labiolingual basal diameter or mesiodistal basal diameter ?

Lines 230-231: with regards to “The premaxillary teeth appear to lack serrations on both carinae of the tooth crown” I don’t see any carinae on those teeth in figure 1. If indeed present, please describe the carinae and then mention that they lack serrations.

Lines 236-238: describe the surficial texture of the premaxillary teeth here and then describe the surficial texture of the anterior dentary and maxillary dentition as consistent with that found in the premaxillary dentition.

Line 247: the parenthatical statement “vertical fluting to form an effective piercing edge” is interpretive and belongs not in the desrciption section but the discussion section. Also, note that “fluting” actually comprises long narrow furrows with rounded bottoms, whereas the term “reeding” is used for (more or less) parallel ridges with rounded surfaces (e.g. google “Fluted vs Reeded” for visual examples). the ridges on the teeth shown in figure 2 are better described as “reeding” to me.

Line 260: explanation for abbreviation “c” (carina [?]) missing.

Line 267: change “tooth position mx05, mx08, mx10, mx14” to “tooth positions 5, 8, 10, and 14”, Do similar for “mx15 or mx16” in next line and line 270. Label these tooth numbers on figure 3.

Lines 282-283: NOTE THAT the left maxilla (tooth count of 13) is shown in parts A and B and the right maxilla (tooth count of 24) is shown in parts C and D (the reverse is indicated in the caption). NOTE THAT part A and C are lateral views and parts B and D are dorsal views. Also, please rotate image in part D 180 degrees to facilitate comparison (lining up) with C.

Lines 285-288: change these lines to “Although incomplete, the right jugal (Fig 4) exhibits the triradiate shape typical of early diapsids. The dorsal process is complete and displays a characteristic flat spoon shape up to the tip where the anterior surface would suture to the postorbital bone.”

Line 297: typo in “subtermporal”

Line 304: change “than in the neodiapsid” to “that of the neodiapsid”

Line 314: change “while the left pterygoid preserves only its central palatal portion” to “whereas only the central palatal portion of the left pterygoid is preserved.”

Line 317: change “lateral most” to “lateral-most”

Line 323: “but to a smaller degree” [?]

Line 330: change “shelves” to “low mounds” or “swellings” [?]

Line 336: please label the ectopterygoid in figure 5 given that it is “not shown as a separate element in segmentation”

Lines 338-347: if only “the medial anterior portion of the left palatine is present” how does the “lateral process indicate “the presence of a rather large suborbital fenestra” ?

Line 349: the cultriform process is trough shaped and only “U-shaped” in cross section.

Line 353: change “foramen” to “foramina”.

Line 362: change “right mandible” to “right ramus”

Line 363: change “left mandible” to “left ramus”

Line 366: delete “elements”

Line 404: change “other parareptiles or eureptiles” to “other eureptiles or in parareptiles”

Line 415: capitalize “carboniferous”

Lines 418-423: change the tooth position “nomenclature” from “d01 to d03” to “1 to 3”, etc. in this paragraph. Do the same for lines 440 and 511.

Lines 419-420: with regards to “with the largest teeth observed in mid-section (d05 to d08)” does not make sense if there are 21 dentary teeth; the “mid-section: would be, if anything, comprised of dentary teeth 8 to 12. It looks to me that dentary teeth 5 to 8 form a caniniform region.

Lines 285-XXX: change these lines to “ROMVP 87367, an additional fragmentary right dentary belonging to another individual of Maiothisavros diane, was also segmented (Fig 8).”

Line 441: change “Abbreviations:” to “Abbreviation:”

Lines 448-450: Malcolm Heaton (1979) identified this foramen as the foramen intermandibularis oralis. You can see it quite clearly in the left splenial in figure 6b (hint: you should label it). Finally, reword the sentence beginning “The internal or lateral surface contains the same single large foramen. . .”

Line 479: change “left mandible” to “left mandibular ramus”

Line 534: change “Abbreviations:” to “Abbreviation:”

Line 539: change “early Permian” to “lower Permian”

Line 589: typo in “phylogenic”

Line 593: change “synapsids canal has” to “canals in these synapsids have”

Line 614: close up “sub orbital” and italicize “O. mayorum”

Line 634: what is the base of the tooth? Is this a term used in lieu of root because the authors regard the implantation to be pleurodont? It looks to me that the plicidentine folds are not exposed because they are covered by bone of attachment, as in Captorhinus aguti (see de Ricqles Bolt 1983).

Line 636: what is “lamalle” ? Lamella? Lamellae?

Line 644: change “Mosaurus” to “Mosasaurus” [?] and italicize

Line 693: italicize “Petrolacosaurus”

Line 699: change “Ford, Benson, R. B. J.” to “Ford, D. P., Benson, R. B. J.” and italicize “Orovenator mayorum”

Line 702: change “Debraga” to “deBraga”

Line 717: change “Belly” to “Bally”

Line 731: italicize “Alligator”

Line 733: italicize “Anguis fragilis”

Line 742: italicize “Deloyhynchus”

Line 756: italicize “Dinilysia”

Figure 5: label the margin of the palatine that contributes to the suborbital fenestra. This is critical for the identification of the specimen as diapsid.

REFERENCE

LeBlanc, A.R.H., et al. 2020. Tooth attachment and pleurodont implantation in lizards: Histology, development, and evolution. J. of Anatomy 238:1156-1178.

Reviewer #3: OVERALL COMMENTS -

This manuscript describes an exceptional new taxon of amniote from the Richard's Spur locality of Oklahoma. The authors argue, I think accurately, that the new taxon is a member of Diapsida. My review comments are restricted to a general issue with some of the specifics of the phylogenetic discussion and individual comments on the description. Numerous times in the text, the authors suggest that the new taxon may not only be a diapsid, but a neodiapsid, a more inclusive group of animals more crown-ward relative to Araeoscelidea. To further explore that concept, I think the paper would be aided by the inclusion of the new taxon in a phylogenetic study of early amniotes. Problematically, an analyitical framework with the range of taxa required to asssess this taxon are very limited. I might suggest the following:

The original analysis used to assess Orovenator mayorum by Reisz et al. (2011). Although limited in species diversity, it does properly capture the character transitions described in the paper that support the position of the new taxon as a diapsid and possibly a neodiapsid.

The analysis of Ford et al. (2020) exploring the phylogeny of Amniota. Although the recovery of Araeoscelidia as well removed from Neodiapsid remains controversial, including among authors on this submissison, it may be interesting to see how the new taxon is recovered here.

The analysis of Schoch and Sues (2018) examining the phylogenetic position of the early lepidosauromorph Fraxinisaura. Although that taxon is well removed from the animal described in this submission, the phylogenetic matrix covers a wide range of early amniotes, ranging from captorhinomorphs to “pelycosaurs” to early diapsids.

I present my individual comments on the paper below. The figures are all excellent and capture the morphology described in the text well. I have only a few suggestions for additional call-outs in the comments below.

LINE-BY-LINE EDITS -

Page 1

Line 27 – Are you intending the “both lower and upper temporal fenestrae” to refer to all the araeoscelideans and neodiapsid? It is confusing as stated, and troublesome considering the closed fenestra in Araeoscelis and the unknown conditions in more incomplete taxa.

Page 3

Line 53 - Seems like these two sentences should be divided. Both points are valid, but do not flow together.

Line 55 – represent

Line 56 – Recommend citation of Sues' recent book "The Rise of REptiles" or Carroll's "Rise of Amphibians." Basically any book titled "the Rise of..."

Line 58 – Birds and crocodylians should receive mention here as well.

Lines 61-62 - I would emphasize this refers primarily to species-level diversity rather than morphological disparity, unless you mean the latter or both.

Page 4

Line 69 – Cite a bunch of younginiform, weigeltisaurid papers here.

Page 5

Line 101 – I think you should note that neutron tomography was used here. Quite important to distinguish from usual microCT.

Page 9

Lines 194-199 - Should the fluted tooth crowns be noted here as well? That does not seem present in other Palaeozoic diapsids. Adds to a good combination of characters.

Page 10

Line 213 – slender relative to a specific group of other diapsids or general amniotes?

Line 215 – a citation to extant predators using this approach would support the statement.

Line 216 – Killed how?

Line 228 - "single space" is vague here. Are you referring to a space equivalent to the mesiodistal width of the other premax teeth?

Line 229 - More than what?

Line 230 - Personal preference, but "labiolingually" is commonly used for tooth dimensions.

Page 11

Line 232 - posteriormost or distalmost.

Line 233 - "conforms" might be better. "Fits" calls to mind the association of the upper and lower teeth or bones.

Line 234 - A bit vague. The base of the crown? Root?

Line 237 - As this is the first major reference to the crown texture, a description (at least a brief one) would be worthwhile for readers.

Line 239 - narrow and elongated in which axes? Sorry, I'm really picky on these descriptors.

Line 241 – Broken. Determined because of cracks on the elements? How do you know?

Line 244 – Reference that your argument in favor of the presence of plicidentine will be provided below in the tooth section.

Line 245 - Indicate which tooth is the largest. It is a pretty common early amniote/diapsid thing to have some bigger "caniniform" teeth towards the front of the max. This is not present in later neodiapsids like Youngina and Claudiosaurus.

Line 246 - Any chance this groove could be indicated in Figure 2?

Line 250 - Could this also be a consequence of the larger root of the largest maxillary tooth?

Page 12

Line 253 – Indicate both of these in the figure. The suborbital contribution is especially hard to interpret as-is.

Line 265 - the long axis of the maxilla?

Page 13

Line 286 - To be more explicit, "anteriorly concave spoon-like shape?"

Lines 292-293 - Somewhat vague about the jugal anterior process. Perhaps "is broken anteriorly."

Page 14 -

Line 316-317 – The size of these pterygoid TP teeth is an interesting character that might be added to the diagnosis. Not sure how widely distributed transverse process teeth on that scale are in Diapsida.

Page 15 -

Line 336 - You might note the absence of such ectopterygoid teeth among neodiapsids.

Page 16 -

Line 380 - They're quite separate phylogenetically, but you might make comparisons with Tanystropheus and Azendohsaurus, regarding the expanded symphyseal region.

Page 17 -

Line 390 - Perceived homology with the Meckelian groove?

Lines 405-406 - A condition converged upon by some later archosauromorphs, such as Macrocnemus (https://www.nature.com/articles/s41598-020-68912-4).

Page 18 -

Lines 410-416 - Comparisons with pathologes in mesozoic or cenozoic diapsids would be ideal here, too.

Line 417 - Definitions for these conditions are so hyper-variable that a citation for the specific criteria would be helpful.

Page 21 -

Line 498 – Is there a specific way the acellularity of the tissue was determined?

Page 23 -

Line 541-542 – Does this refer specifically to the Permian of Oklahoma or something broader? Confusing as written.

Page 25 -

Line 576-577 - You might cite an early amniote phylogeny that supports the suborbital fenestra as a diapsid character.

Line 590 – Regarding this maxillary innervation character, you might note that more information on the plesiomorphic diapsid conditions and that of early lepidosauromorphs will be crucial to resolving this. Still, it's a VERY interesting hypothesis.

Page 26 -

Line 614 – Italicize O. mayorum.

Page 28 -

Line 644 – Mosasaurs?

Page 29 -

Line 668 – A few references to the new taxon possibly being a neodiapsid argue, in my opinion, for the inclusion of a phylogenetic analysis for the manuscript. I don't see it as an absolute necessity, but I think the neodiapsid references might best be removed otherwise.

Lines 673-675 – Odd to argue for a phylogenetic analysis in a paper that lacks such an analysis. You may indicate whether a follow-up is coming.

Line 674 – phylogenetic.

Pages 30-33 -

There are many spelling errors, formatting issues, and italics problems through the citations. Please revise.

6. PLOS authors have the option to publish the peer review history of their article (what does this mean?). If published, this will include your full peer review and any attached files.

Reviewer #1: **Yes: **David P. Ford

Reviewer #2: **Yes: **Sean Modesto

Reviewer #3: **Yes: **Adam Pritchard

---

## [Author Response · Author response to Decision Letter 0]

24 Sep 2022

We have made all the modification requested by the reviewers and the editor. We have also attached a detailed series of comments to each of the requested modifications in the uploaded files. Finally, we have activated online availability of CT data on Morphobank as requested.https://morphobank.org/index.php/Projects/ProjectOverview/project_id/4395

---

## [Editor Report · Decision Letter 1]

6 Oct 2022

PONE-D-22-17394R1An intriguing new diapsid reptile with evidence of mandibulo-dental pathology from the early Permian of Oklahoma revealed by neutron tomographyPLOS ONE

Dear Dr. Reisz,

Thank you for submitting your manuscript to PLOS ONE. After careful consideration, we feel that it has merit but does not fully meet PLOS ONE’s publication criteria as it currently stands. Therefore, we invite you to submit a revised version of the manuscript that addresses the points raised during the review process. It is almost done. You need fix a few minor errors before it is accepted. 

We look forward to receiving your revised manuscript.

Kind regards,

Jun Liu

Academic Editor

PLOS ONE

Journal Requirements:

Additional Editor Comments :

L248 posterior-most  ?

L272: 5 to Five

Ref: 32 the journal name is missing

---

## [Editor Report · Decision Letter 2]

14 Oct 2022

An intriguing new diapsid reptile with evidence of mandibulo-dental pathology from the early Permian of Oklahoma revealed by neutron tomography

PONE-D-22-17394R2

Dear Dr. Reisz,

We’re pleased to inform you that your manuscript has been judged scientifically suitable for publication and will be formally accepted for publication once it meets all outstanding technical requirements.

Kind regards,

Jun Liu

Academic Editor

PLOS ONE
---

## [Editor Report · Acceptance letter]

9 Nov 2022

PONE-D-22-17394R2 

An intriguing new diapsid reptile with evidence of mandibulo-dental pathology from the early Permian of Oklahoma revealed by neutron tomography. 

Dear Dr. Reisz:

I'm pleased to inform you that your manuscript has been deemed suitable for publication in PLOS ONE. Congratulations! Your manuscript is now with our production department. 

Kind regards, 

on behalf of

Dr. Jun Liu 

Academic Editor

PLOS ONE